# LOCRET: ENHANCING EVICTION IN LONG-CONTEXT LLM INFERENCE WITH TRAINED RETAINING HEADS

## ABSTRACT

Large language models (LLMs) have shown remarkable advances in supporting long-context comprehension and processing tasks. However, scaling the generation inference of LLMs to such long contexts incurs significant additional computation load, and demands a substantial GPU memory footprint to maintain the key-value (KV) cache of transformer-based LLMs. Existing KV cache compression methods, such as quantization, face memory bottlenecks as context length increases, while static-sized caches, such as selective eviction, suffer from inefficient policies. These limitations restrict deployment on consumer-grade devices like a single Nvidia 4090 GPU. To overcome this, we propose LOCRET, an efficient framework for long-context LLM inference that introduces *retaining heads* to evaluate the causal importance of KV cache units, allowing for more accurate eviction within a fixed cache size. LOCRET is fine-tuned on top of the frozen backbone LLM using a minimal amount of data from standard long-context SFT datasets. During inference, we evict low-importance cache units along with a chunked prefill pattern, significantly reducing peak GPU memory usage. We conduct an extensive empirical study to evaluate LOCRET, where the experimental results show that LOCRET outperforms the recent popular and competitive approaches, including INFLLM, Quantization, SIRLLM, and MINFERENCE, in terms of memory efficiency and the quality of generated contents — LOCRET achieves over a $20\times$ and $8\times$ KV cache compression ratio compared to the full KV cache for `Phi-3-mini-128K` and `Llama-3.1-8B-instruct`. Additionally, LOCRET can be combined with other efficient inference methods, such as quantization and token merging. To the best of our knowledge, LOCRET is the first framework capable of deploying `Llama-3.1-8B` or similar models on a single Nvidia 4090 GPU, enabling 128K long-context inference without compromising generation quality, and requiring little additional system optimizations.

## 1 INTRODUCTION

Large language models (LLMs) have revolutionized AI development and deployment (Zhao et al., 2023; Minaee et al., 2024). Recent advancements in LLMs' ability to handle long-context tasks have further unlocked the potential of generative AI. State-of-the-art LLMs now support significantly extended context lengths, with GPT-4o (OpenAI, 2024) and Llama-3.1 (Dubey et al., 2024) handling 128K tokens, Yi (Young et al., 2024) and Claude-3 (Anthropic, 2024) supporting 200K tokens, and Gemini-1.5 (Reid et al., 2024) reaching 10 million tokens. These advances enable LLMs to tackle complex tasks like multi-hop reasoning (Li et al., 2024a; Schnitzler et al., 2024), solving needle-in-a-haystack problems (Guerreiro et al., 2023; Wang et al., 2024a), and powering advanced LLM agents (Qin et al., 2023; Wang et al., 2024b) and AI-driven operating systems (Mei et al., 2024). However, *deploying generative inference under long-context settings on consumer-grade GPUs requires innovative algorithmic and system optimizations to handle this new paradigm efficiently.*

Compared to traditional short-context LLM inference, long-context LLM inference shifts the computing paradigm in two key ways: i) *increased computational overhead for attention mechanisms*: as context length grows, the computation required for obtaining attention scores increases quadratically, which results in a higher ratio of the computational budget in a transformer block; ii) *higher memory footprint for key-value (KV) caching*: longer contexts require larger KV caches, which dra-

matically increases the peak memory usage. These shifts demand innovative techniques to mitigate computational costs and manage memory usage effectively for long-context LLM inference.

Although various efforts have been made to overcome the bottleneck in LLM inference, these approaches fail to enable long-context inference on consumer-grade GPUs. Models with compact architectures, e.g. `MiniCPM-128K` (Hu et al., 2024b) and `Phi-3-mini-128K` (Abdin et al., 2024), reduce the computational load and memory usage, but cannot alleviate the KV cache burden in long-context scenarios. Similarly, techniques like LLM model weight quantization (Frantar et al., 2023; Lin et al., 2024; Ma et al., 2024), activation quantization (Dettmers et al., 2022; Xiao et al., 2023; Zhang et al., 2024c), or sparsification (Liu et al., 2023; Zhang et al., 2022) also fall short in effectively reducing the memory usage to a level supported by consumer-grade GPUs.

Recently, some specific optimizations have been proposed for long-context LLM inference. For example, sparse attention mechanisms (Jiang et al., 2024a; Ge et al., 2024; Lou et al., 2024) attempt to reduce runtime memory through conduct selected few calculation, and KV cache quantization (Liu et al., 2024b; Hooper et al., 2024; Zandieh et al., 2024) reduces cache size by applying low-bit storage. These methods can only offer limited compression rate, of which the core issue is that *the KV cache grows linearly with sequence length*, and the above methods do not adequately address this problem. On the contrary, combining chunked prefill with token-dropping techniques (Xiao et al., 2024b; Yang et al., 2024) could offer a more effective solution, as it maintains a static-sized cache where the memory usage can be bounded. However, current token-dropping and cache eviction methods (Zhang et al., 2024e; Liu et al., 2024a; Yao et al., 2024), whose token importance is manually designed according to the inner statistics, suffer from accuracy loss and performance degradation due to inaccuracies in estimating token importance — The weakening correlation between local and global importance as sequences grow exacerbates this issue. Existing scoring functions of token importance, e.g. $H_2O$ (Zhang et al., 2024e) and SNAPKV (Li et al., 2024b), utilize the information of the subsequent tokens, making them incompatible to the chunked prefill pattern. Other scoring functions that do not use the subsequent information, like SIRLLM (Yao et al., 2024), exhibit significant performance degradation. Here, we visualize the consistency of the top-10% cache unit labeling among different scoring functions to show the weakening correlation in Figure 1, and more details are elaborated in Appendix B. To address these limitations, we propose a *lightweight training-based paradigm* that provides more accurate token importance scoring to tackle the long-context LLM inference problem. We highlight our contributions below:

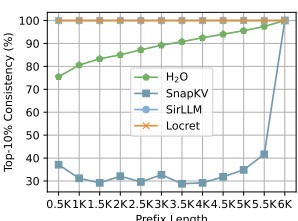

Figure 1: The Top-10% Consistency of typical cache importance scoring functions.

**Contribution 1:** We propose a lightweight training-based paradigm for selective KV cache eviction for long-context LLM inference, with an offline training cost <1 GPU hours. We tackle the problem of KV cache eviction by a learning based approach. We introduce the *retaining heads*, with a small number of additional parameters, fine-tuned on top of the frozen backbone LLM using a minimal amount of data from standard long-context SFT datasets to estimate the causal importance of each cache unit. Such a training paradigm is able to provide accurate token importance scoring prediction and can be integrated with other efficient inference algorithms, e.g., quantization and token merging.

**Contribution 2:** We provide an efficient inference system implementation for LOCRET. We integrate the retaining head mechanism into a chunked prefill inference framework, where we maintain a static-size cache set through evicting cache units with low predicted importance to limit the GPU memory usage. LOCRET is able to preserve the most important cache units with the trained retaining heads, enabling precise attention approximation without compromising the inference latency. LOCRET is also applicable to all transformer-based LLMs and various hardware, as it requires minimal modifications to the model's inference process and only utilizes dense operators.

**Contribution 3:** We conduct an extensive evaluation of LOCRET, which illustrates that LOCRET can not only obtain a comparable performance but also maintain inference efficiency. LOCRET achieves over a $20\times$ and $8\times$ KV cache compression ratio for `Phi-3-mini-128K` and `Llama-3.1-8B-instruct`, enabling full comprehension of long contexts on consumer-grade devices. To the best of our knowledge, LOCRET is the first framework capable of deploying `Llama-3.1-8B` or similar models on a single Nvidia 4090 GPU, enabling 128K long-context inference without compromising generation quality, and requiring little extra system optimizations.

## 2 RELATED WORK

Efforts in long-context LLM inference can be categorized by algorithm and system optimizations:

**Algorithm optimizations.** Optimizations aimed at reducing the size of the KV cache can generally be classified into three categories: quantization-based methods, token dropping methods, and sparsity-based methods. Quantization-based methods (Liu et al., 2024b; Hooper et al., 2024; Zandieh et al., 2024; Zhang et al., 2024a), which store the KV cache in low-bit representations, require hardware support for these formats and may slow down inference due to the overhead of dequantization. Token dropping methods typically follow two main strategies: eviction or the use of an attention pool. Eviction-based approaches, such as $H_2O$ (Zhang et al., 2024e), ScissorHands (Liu et al., 2024a), and SIRLLM (Yao et al., 2024), rank tokens by certain statistical metrics to identify the most influential ones, discarding others to reduce memory usage. Attention pool-based methods (Nawrot et al., 2024; Rajput et al., 2024), such as StreamingLLM (Xiao et al., 2024b) and LoCoCo (Cai et al., 2024a), compress multiple adjacent KV cache units into a single unit using a specially designed transformation. Sparsity-based methods (Ge et al., 2024; Jiang et al., 2024a; Yang et al., 2024; Lou et al., 2024; Lv et al., 2024) focus on leveraging the sparsity patterns of attention heads to reduce both computation and I/O. The combination of these approaches can be further enhanced by identifying specific attention patterns for each head and layer (Ge et al., 2024). For surveys of these methods, please refer to (Yuan et al., 2024; Kang et al., 2024; Shi et al., 2024).

**System optimizations.** The challenge of long-context inference can also be alleviated from a system-level perspective. Offloading-based methods (Sheng et al., 2023; Xiao et al., 2024a; Wu et al., 2024; Sun et al., 2024) store the KV cache in CPU memory, retrieving only the most relevant chunks to the GPU before computing a new chunk. This approach reduces peak GPU memory usage, though at the cost of slower inference. Hardware-aware algorithms, such as flash attention (Dao et al., 2022; Dao, 2024; Shah et al., 2024) and page attention (Kwon et al., 2023), exploit GPU architecture (Ghorpade et al., 2012) to enable more efficient runtime memory management. In addition, reimplementing inference infra-structure in a more efficient programming language (llama.cpp; llama2.c; rustformers), or adopting disaggregated inference (Jiang et al., 2024b; Zhong et al., 2024; Qin et al., 2024; Hu et al., 2024a), can greatly enhance inference efficiency. Algorithmic optimizations can be seamlessly integrated into such systems (Agrawal et al., 2023; Lee et al., 2024). For instance, KTransformers (KVCache.AI, 2024) adopts the chunked offloading technique from INFLLM (Xiao et al., 2024a). However, system optimizations primarily focus on extending context length by leveraging hardware resources, rather than directly reducing the size of the KV cache.

## 3 LOCRET

### 3.1 PRELIMINARIES

**Transformer architecture.** We define the model inference of transformer-based LLMs as follows. Given a token sequence $t_1, t_2, \cdots, t_n$, we denote the output hidden state of layer $i$ as $\mathbf{H}^{(i)}$ and $\mathbf{H}^{(0)}$ is the embeddings. Each transformer layer consists of an attention layer and an MLP layer. We assume the model follows a grouped-query attention (GQA) architecture (Ainslie et al., 2023), with $h$ query heads and a group size of $g$. For multi-head attention (MHA), $g$ is set to 1. The attention score of layer $i$'s head $j$ is calculated by $\mathbf{A}_j^{(i)} = \text{softmax}\left(\mathbf{Q}_j^{(i)}\mathbf{K}_{\lceil j/g \rceil}^{(i)T}/\sqrt{d_m}\right) \cdot \mathbf{V}_{\lceil j/g \rceil}^{(i)}$, where $d_m$ represents the hidden size for each head and $\left[\mathbf{Q}_j^{(i)}, \mathbf{K}_j^{(i)}, \mathbf{V}_j^{(i)}\right] = \mathbf{H}^{(i-1)} \cdot \left[\mathbf{W}_j^{Q(i)}, \mathbf{W}_j^{K(i)}, \mathbf{W}_j^{V(i)}\right]$. Next, we compute $\mathbf{A}^{(i)} = \left[\mathbf{A}_1^{(i)}, \cdots, \mathbf{A}_h^{(i)}\right] \cdot \mathbf{W}^{O(i)}$, finally followed by $\mathbf{H}^{(i)} = \text{MLP}(\mathbf{A}^{(i)})$.

**KV cache and chunked prefill.** During the prefill stage, all prompt tokens are processed in a single forward pass, where $\mathbf{Q}^{(i)}$, $\mathbf{K}^{(i)}$, and $\mathbf{V}^{(i)}$ each have a sequence length of $n$. In the decoding stage, only a single token is processed across layers, utilizing the KV cache units to reduce computation. Chunked prefill is a method for reducing peak memory consumption by processing tokens in chunks over multiple passes, with the assistance of the KV cache. Taking both KV cache and chunked prefill into account, the attention calculation can be modified as Equation 1, where $B$ represents the number of tokens processed in a single model pass. For decoding, $B = 1$, while for chunked prefill, $B$ corresponds to the chunk size. We denote the attention output for tokens $n+1, n+2, \cdots, n+B$

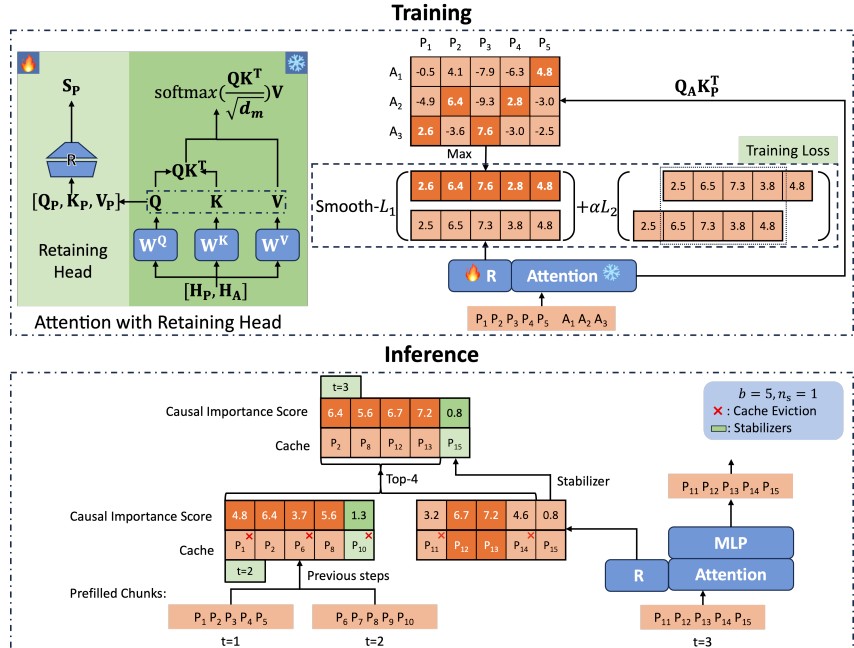

Figure 2: The framework of LOCRET. "$\mathbf{R}$" represents the retaining head. $P_i$ and $A_i$ correspond to the $i$-th prompt token and answer token. "t" represents the time step in chunked prefill, "$b$" represents the budget size, and "$n_s$" represents the length of the stabilizers.

as $\mathbf{A}[n+1\colon n+B]$, and the attention output for the $k$-th token as $\mathbf{A}[k]$.

$$\mathbf{A}[n+1\colon n+B]_j^{(i)} = \text{softmax}\left(\frac{\mathbf{Q}[n+1\colon n+B]_j^{(i)}\mathbf{K}[1\colon n+B]_{\lceil j/g\rceil}^{(i)T}}{\sqrt{d_m}}\right) \cdot \mathbf{V}[1\colon n+B]_{\lceil j/g\rceil}^{(i)}. \quad (1)$$

**Cache eviction.** Cache eviction in long-context inference is defined as follows. Here, we slightly abuse the notation of heads and layers, and treat the key-value vector pair of a single token within one head as the smallest cache unit. We denote the cache unit for the $k$-th token as $c_k = (\mathbf{K}[k], \mathbf{V}[k])$. Assume a memory budget $b$, representing the maximum number of cache units that can be stored in GPU memory at any given time. The abstract form of attention can then be written as $c_k = f(c_1, c_2, \cdots, c_{k-1})$. With limited cache capacity, this calculation can only be approximated by $\tilde{c}_k = f(\tilde{c}_{p_1}, \tilde{c}_{p_2}, \cdots, \tilde{c}_{p_{b'}})$, where $b' \leq b$, and $p_1, p_2 \cdots, p_{b'} \in \{1, 2, \cdots, k-1\}$. Intuitively, the number of prior cache units involved cannot exceed the memory budget. When $b' = b$, indicating the cache is full, one cache unit must be evicted. We select the unit to be evicted using a policy $p_v = \text{Policy}(\tilde{c}_{p_1}, \cdots, \tilde{c}_{p_b}; \tilde{c}_k)$. In such stated problem, the key challenge of cache eviction is to develop an effective policy function that minimizes the approximation error $\|\tilde{c}_k - c_k\|$.

## 3.2 LOCRET FRAMEWORK

LOCRET is a training-based KV cache compression framework that works in conjunction with chunked prefill. As illustrated in Figure 2, LOCRET operates in two stages: training and inference. In the training stage, we modify the original LLM by appending a retaining head $\mathbf{R}$ to each attention module. We then train the retaining heads $\mathbf{R}$ while keeping the LLM backbone frozen. During chunked prefill inference, the retaining heads $\mathbf{R}$ are used to calculate the importance of each cache unit in the chunk. We retain the cache units with higher scores, along with stabilizers (i.e., the last tokens), in the cache pool located in GPU memory. Through this process, the retaining heads $\mathbf{R}$ learn and predict the patterns discovered by existing methods, as detailed in Appendix L.

From a mathematical perspective, cache eviction is performed by assigning each cache unit an importance score that reflects its influence on the calculation of subsequent tokens. We refer to this estimation as the *causal importance score (CIS)* since it is computed in a causal manner. The CIS of cache unit $k$ is calculated as $s_k = S(c_1, c_2, \cdots, c_k)$. By applying top-$b$ sparse attention based on

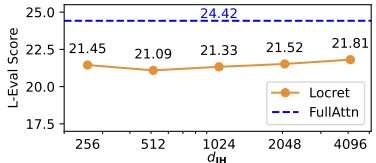

Figure 3: L-Eval scores with different intermediate size of retaining head $d_{\mathbf{R}}$.

Table 1: L-Eval scores of LOCRET trained on various datasets.

| Dataset | LongAlpaca | LongAlign | Anti-Haystack |
|---------|-----------|-----------|---------------|
| L-Eval | 21.33 | 22.00 | 20.72 |

the CIS, we can ensure that the trace (i.e. retaining and eviction) of each cache unit can be fit within a cache with a given memory budget. Further details can be found in Appendix K.

However, since not all tokens can be stored in the cache simultaneously, calculating the actual CIS on-chip is impractical. Instead, we use a heuristic approximation for CIS, defined as follows: $\tilde{s}_k = S(\tilde{c}_{p_1}, \tilde{c}_{p_2}, \cdots, \tilde{c}_{p_{b'}})$, where $b$ is the cache budget, all $\tilde{c}_{p_i}$ are approximated cached units, and $b' \leq b$. We hypothesize that if the scoring function for causal importance is sufficiently accurate, it will consistently select the most critical cache units, resulting in a negligible difference between the heuristic and the actual score. Thus, we use the terms *heuristic CIS* and *actual CIS* interchangeably.

### 3.3 TRAINING THE RETAINING HEADS

In this section, we introduce LOCRET's model architecture modifications and the corresponding training recipe. We add additional parameters to compute the CIS $s_k$ (or $\tilde{s}_k$ for on-chip inference) with respect to all previous cache units. Specifically, we inject a retaining head, consisting of a small MLP, into each layer. From our observation, such small MLPs do not slow down model inference, with details elaborated in Appendix J. The retaining head predicts the CIS for each head of the corresponding layer based on the concatenation of $[\mathbf{Q}, \mathbf{K}, \mathbf{V}]$. Formally, with a slight abuse of notation, let the retaining head for layer $i$ be denoted as $\mathbf{R}$. The CIS at head $j$ of layer $i$ is then calculated as: $\tilde{\mathbf{S}} = \mathbf{R}([\mathbf{Q}, \mathbf{K}, \mathbf{V}]) = \sigma([\mathbf{Q}, \mathbf{K}, \mathbf{V}]\mathbf{W}_1)\mathbf{W}_2$. In this equation, $\mathbf{W}_1 \in \mathbb{R}^{(d_m + 2d_{kv}) \times d_{\mathbf{R}}}$ and $\mathbf{W}_2 \in \mathbb{R}^{d_{\mathbf{R}} \times \frac{h}{g}}$ are the tunable parameters of $\mathbf{R}$, $\sigma$ is the activation function and $\tilde{\mathbf{S}}[k]_j$ is the predicted CIS of the $k$-th token at head $j$ of layer $i$. This architecture implies that the importance estimation for a single head is not performed in isolation but rather considers all heads together. Note that for GQA models, there are only $h/g$ output values corresponding to the number of heads in the KV cache.

We train the retaining head $\mathbf{R}$s on a small Question-Answer (QA) supervised fine-tuning (SFT) dataset, where each entry consists of a single prompt and one answer. We define the CIS $s_k$ for the $k$-th token as the maximum attention score, before softmax, from all the answer tokens toward the $k$-th token. Formally, for the $k$-th token at head $j$ of layer $i$, we approximate the predicted value $\tilde{\mathbf{S}}[k]_j^{(i)}$ to the ground truth $\mathbf{S}[k]_j^{(i)} := \max_p \left( \mathbf{Q}_j^{(i)} \mathbf{K}_j^{(i)T} \right)_{p,k}$, where $n_q(d) \leq p \leq n_q(d) + n_a(d)$, and $n_q(d)$ and $n_a(d)$ represent the lengths of the prompt and answer in data $d$, respectively. For an MHA model with $L$ layers and $h$ heads, the training objective is described in Equation 2. For GQA models, we take the maximum attention score before softmax across different query heads within the same group as the ground truth for the corresponding KV head.

$$\underset{\mathbf{W}_1^{(i)}, \mathbf{W}_2^{(i)}, i=1,2\cdots,L}{\arg\min} \mathbb{E}_{d \in \mathcal{D}} \left[ \sum_{i=1}^{L} \sum_{j=1}^{h} \sum_{k=1}^{n_q(d)} \mathcal{L}\left( \tilde{\mathbf{S}}[k]_j^{(i)}, \mathbf{S}[k]_j^{(i)} \right) \right] \tag{2}$$

The training loss consists of a regression loss and a smoothing loss. We apply the Smooth-$\mathcal{L}_1$ norm between the predicted values and the ground truth. Since important segments in natural language typically consist of adjacent tokens, we also apply the $\mathcal{L}_2$ norm between each pair of adjacent predicted values to enforce smoothness. The complete training loss for LOCRET is given by Equation 3.

$$\mathcal{L}\left( \tilde{\mathbf{S}}[k]_j^{(i)}, \mathbf{S}[k]_j^{(i)} \right) = \text{Smooth-}\mathcal{L}_1\left( \tilde{\mathbf{S}}[k]_j^{(i)}, \mathbf{S}[k]_j^{(i)} \right) + \alpha \mathcal{L}_2\left( \tilde{\mathbf{S}}[k]_j^{(i)}, \tilde{\mathbf{S}}[k+1]_j^{(i)} \right) \tag{3}$$

From our observations, the training of LOCRET exhibits strong robustness. The performance variations shown in Figure 3 and Table 1 are minimal, despite changes in $d_{\mathbf{R}}$ and the dataset. Details can be found in Appendix F. Training statistics, including loss dynamics, are recorded in Appendix M.

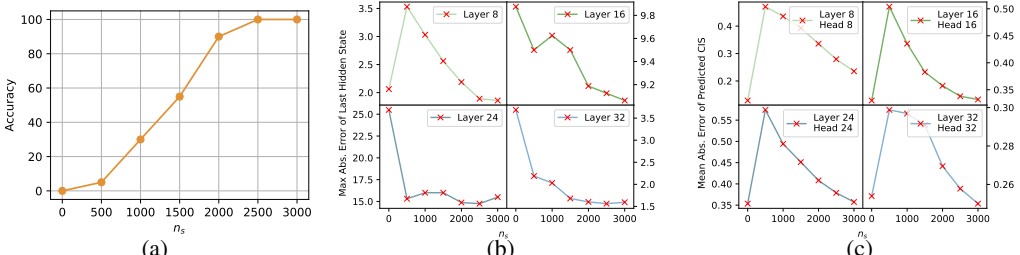

Figure 4: R.Number with different stabilizer lengths $n_s$. (a) Task accuracy under different $n_s$. (b) Maximum absolute error of the last hidden state. (c) Mean absolute error of the predicted CIS. We conduct this experiment on entries 101-120 of R.Number using the `Phi-3-mini-128K` backbone.

---

**Algorithm 1:** LOCRET Inference

---

**Input:** Model $\mathbf{M}$, Prompt tokens $x$, Local length $n_{loc}$, Stablizer length $n_s$, Budget $b$, Chunk size $B$
**Output:** Generated tokens $x_{gen}$
```
// Leave the last n_loc out to make sure they are not evicted.
```
chunk_positions $\leftarrow$ split_chunk(0, $x$.length() $-n_{loc}$, $B$)
K_cache, V_cache, score_cache $\leftarrow$ [], [], []
**for** chunk $\in$ chunk_positions **do**
    begin_pos, end_pos $\leftarrow$ chunk.begin_pos, chunk.end_pos
    K_chunk, V_chunk, score_chunk $\leftarrow$ $\mathbf{M}(x$[begin_pos:end_pos], K_cache, V_cache)
    K_cache $\leftarrow$ Concat(K_cache, K_chunk)
    V_cache $\leftarrow$ Concat(V_cache, V_chunk)
    score_cache $\leftarrow$ Concat(score_cache, score_chunk)
    **if** chunk is not the last chunk **then**
```
        // Keep the last n_s caches to maintain higher context continuity.
```
        score_cache[score_cache.length()-$n_s$:score_cache.length()] $\leftarrow$ $+\infty$
    **end if**
    indices $\leftarrow$ top-$b$(score_cache).indices
    K_cache, V_cache, score_cache = K_cache[indices], V_cache[indices], score_cache[indices]
**end for**
K_cache, V_cache, score_cache $\leftarrow$ $\mathbf{M}(x$[$x$.length()$-n_{loc}$:$x$.length()], K_cache, V_cache)
$x_{gen} \leftarrow \mathbf{M}$.generate(K_cache, V_cache)
**return** $x_{gen}$

---

### 3.4 INFERENCE IMPLEMENTATION OF LOCRET

During the inference stage, we use the chunked prefill pattern and perform cache eviction based on the predicted CIS. Since the predicted value $\tilde{\mathbf{S}}[k]_j^{(i)}$ depends solely on $\mathbf{Q}[k]_j^{(i)}$, $\mathbf{K}[k]_j^{(i)}$, and $\mathbf{V}[k]_j^{(i)}$, and because attention in decoder-only models is causal, $\tilde{\mathbf{S}}[k]_j^{(i)}$ remains consistent once calculated. Thus, we store the KV cache units along with their corresponding causal importance values. When the cache is full, we evict the units with lower causal importance values, as they are deemed less useful for future computations. Cache eviction introduces context discontinuity, meaning some cache units at certain positions may be absent. This can degrade generation quality and increase the error between the predicted and accurate CIS, as LLMs are typically not trained on such contexts. To mitigate this, we retain the last tokens of the current chunk at each step of the chunked prefill process, ensuring a local and continuous context to minimize errors. To demonstrate the effectiveness of this design, we perform an ablation study on the length of stabilizers $n_s$, shown in Figure 4. Smaller $n_s$ results in severe performance degradation, and the model fails entirely when stabilizers are absent, as context discontinuity leads to instability in CIS prediction, causing errors in cache eviction and amplifying errors in hidden states. More details are discussed in Appendix I.

We maintain a cache pool with a capacity of $b$ cache units, discarding units that exceed this limit. For each chunk, the model processes the chunked input tokens alongside the current cache pool. The newly generated KV pairs and their predicted scores are then concatenated with the existing cache. Once the cache pool is full, only the $b$ cache units with the highest CIS values are retained. At each chunked prefill step, except for the final step, we retain the stabilizers, i.e. the last $n_s$ cache units. Additionally, we do not compress the last $n_{loc}$ tokens of the prompt, as they are critical for maintaining high generation quality due to their strong correlation with the query. Finally, the

answer is generated according to the compressed KV cache. Algorithm 1 provides the pseudocode for LOCRET, where we formally describe the LOCRET inference process.

The GPU memory usage during LOCRET inference can be effectively bounded. GPU memory for KV cache storage is limited to $\mathcal{O}(b + n_{loc})$, and the runtime memory usage of the attention mechanism is bounded by $\mathcal{O}(B \times (b + B + n_{loc}))$. For comparison, while processing an input with $n$ tokens, full attention prefill requires $\mathcal{O}(n)$ for KV cache storage and $\mathcal{O}(n^2)$ for runtime memory, whereas chunked prefill requires $\mathcal{O}(n)$ for KV cache storage and $\mathcal{O}(nB)$ for runtime memory consumption.

## 4 EXPERIMENTS

In this section, we present the experiments conducted to evaluate the proposed framework, LOCRET, aiming to address the following questions:

(**Q1**) Can LOCRET obtain better end-to-end task performance compared to popular and competitive long-conetext inference approaches within similar or less peak memory?

(**Q2**) Can LOCRET improve inference speed compared to other approaches?

(**Q3**) What are the characteristics of LOCRET's hyperparameters?

### 4.1 EXPERIMENTAL SETUP

**Models and training dataset.** We evaluate LOCRET on two long-context LLMs: `Phi-3-mini-128K` (Abdin et al., 2024) and `Llama-3.1-8B-instruct` (Dubey et al., 2024). Both models can process up to 128K context tokens, are suitable for deployment on consumer-grade devices, and follow MHA and GQA architectures, respectively. We inject retaining heads $\mathbf{R}$ into each layer, setting the intermediate size $d_{\mathbf{R}}$ to 1024 for both models. The retaining heads are trained on the LongAlpaca dataset (Chen et al., 2024) for 3000 steps , with a 5e-4 learning rate, 10240 sequence length, and $\alpha$ set to 2.5e-3. Training LOCRET is lightweight, with the tunable parameters comprising 8% and 2.5% of the total for the two models, respectively. The complete training process takes 0.47 and 0.80 GPU hours on a single A800 GPU for each corresponding model. Important hyperparameters are listed in Table 2. More details on hyperparameters as well as the system environment, can be found in Appendix A.

Table 2: Hyperparameters in LOCRET's inference stage. "$b$" refers to cache budget, "$B$" refers to chunk size of chunked prefill, "$n_s$" refers to stabilizers length and "$n_{loc}$" refers to local length.

| Model | $b$ | $B$ | $n_s$ | $n_{loc}$ |
|---|---|---|---|---|
| `Phi-3-mini-128K` | 6000 | 3072 | 2500 | 100 |
| `Llama-3.1-8B-instruct` | 16384 | 1024 | 2500 | 100 |

**Benchmarks.** We evaluate LOCRET on selected subsets of $\infty$Bench (Zhang et al., 2024b) and L-Eval (An et al., 2024). For $\infty$Bench, we select R.PassKey, R.Number, E.Sum, E.QA, E.MC, Z.QA, E.Dia, C.Debug, and M.Find. All selected subsets, except Z.QA, have an average length of approximately 100K tokens, while Z.QA has a longer average length of around 2000K tokens. We exclude R.KV because it can be easily handled by calling a Python interpreter. We also exclude C.Run and M.Calc due to their complexity for all methods, including full attention inference. For L-Eval, we filter out all tasks with an average length shorter than 16384 tokens and evaluate on CodeU, NQ, CUAD, NarrativeQA, QMSum, and SPACE. Metrics are reported according to the recommendations of the two frameworks, with further details provided in Appendix A. We also report the peak memory usage, i.e. the average peak memory measured for the first entry of each task in the corresponding dataset, for reference. Apart from the experiments above, we also evaluate LOCRET on extremely long context dataset, R.PassKey with 10 million tokens, in Appendix G. The experimental results under the multi-turn conversation setting are in Appendix H.

**Baselines.** As discussed in Section 2, existing algorithms for memory-efficient long-context inference can be categorized into offloading-based, quantization-based, token-dropping, and sparsity-based methods. For each category, we select one representative method as the baseline. We compare LOCRET against full attention inference (denoted as FullAttn), INFLLM (Xiao et al., 2024a), KV cache quantization (Turganbay, 2024), SIRLLM (Yao et al., 2024), and MINFERENCE (Jiang et al., 2024a). For quantization, we use Hugging Face Quanto (Hugging-Face) implementation, referring to the 2-bit quantization method as HF-2BITS. We omit HF-4BITS and benchmark this combination in Section E. We do not include attention pool-based token-dropping methods in this benchmark, as

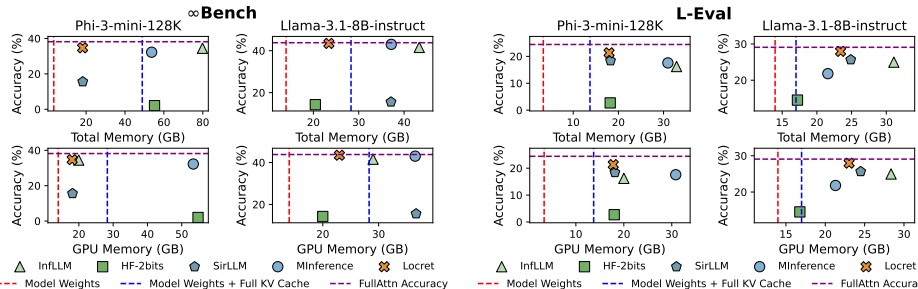

Figure 5: Memory Statistics vs. Task Performance. The red lines correspond to the theoretical size of the model weights, while the blue lines represent the total theoretical size of the model weights and the full KV cache without any compression. The purple lines indicate the accuracies of FullAttn. "Total Memory" represents the total memory usage of both GPU and CPU memory.

they are orthogonal to our approach; further discussion is provided in Section E. Detailed introductions to the selected baselines can be found in Appendix A. We also discuss the comparison between the trained LOCRET and the randomly initialized retaining heads $\mathbf{R}$ in Appendix C.

## 4.2 END-TO-END BENCHMARK

We compare our method with the baselines on both ∞Bench and L-Eval to address **Q1**. As shown in Table 3, LOCRET outperforms all baselines in terms of end-to-end performance.

In the ∞Bench tests, while all methods experience performance degradation compared to FullAttn inference, LOCRET, INFLLM, and MINFERENCE exhibits better performance than other methods, with only a modest drop in performance given the reduced memory usage. Quantization, on the other hand, shows significant degradation and fails on all tasks. SIRLLM performs well on comprehensive tasks such as E.Sum and E.MC, but struggles with tasks that require precise memory, such as R.PassKey and R.Number. LOCRET not only excels in context retrieval tasks but also achieves strong results in comprehensive tasks, earning the highest overall score among all competitors.

In the L-Eval tests, all methods show some degree of performance degradation. Nevertheless, LOCRET achieves the best overall result, obtaining the highest score on most tasks. L-Eval is a shorter but more complex dataset, where SIRLLM performs particularly well. Quantization fails on most tasks, resulting in the lowest overall score. Both INFLLM and MInference suffer significant performance drops compared to FullAttn inference. LOCRET consistently surpasses all competitors.

We also report memory consumption in Figure 5. In the extreme long-context scenario (∞Bench), LOCRET uses relatively less memory while achieving the best overall performance. INFLLM performs well with limited GPU memory usage, but it requires a significant amount of CPU memory to store the full KV cache. Quantization and SIRLLM can achieve low memory consumption in some settings, but quantization introduces severe performance degradation. MINFERENCE employs sparse attention patterns but does not compress the KV cache. As a result, its minimum memory requirement equals the sum of the model weights and the full KV cache. In the shorter context scenario (L-Eval), a similar phenomenon is observed. For `Phi-3-mini-128K`, which has a larger KV cache, INFLLM and MINFERENCE exhibit higher memory consumption due to the need to store the full KV cache. Other methods have similar memory footprints, with LOCRET achieving the best overall performance while using the least memory. For `Llama-3.1-8B-instruct`, whose full KV cache is smaller, the memory bottleneck shifts to runtime computational memory for attention and other calculations. All methods exhibit similar memory footprints, with LOCRET delivering the best overall performance. Our experiments demonstrate that LOCRET is both effective and efficient, outperforming all baselines on multiple datasets and models while using less GPU memory.

## 4.3 SPEED TEST ON REAL CONSUMER-GRADE DEVICES

In this section, we examine the processing speed to demonstrate that LOCRET achieves its strong performance without compromising inference speed, addressing question **Q2**. We evaluate the inference speed on the R.PassKey task from ∞Bench and compare LOCRET against all the baselines introduced in Section 4.1, using a single Nvidia 4090 GPU with 24GB of memory, which is typical for consumer-grade AI devices. We report the inference speed as the total number of tokens in the

Table 3: The experimental results of LOCRET compared with all the baselines on ∞Bench and L-Eval, where higher score represents better performance. "Avg." represents the average score across all tasks. The highest score in each column is marked in **bold**, and the second highest is underlined. LOCRET achieves the highest overall score among all competitors in every setting.

| Method | R.PassKey | R.Number | E.Sum | E.QA | E.MC | Z.QA | E.Dia | C.Debug | M.Find | Avg.↑ |
|---|---|---|---|---|---|---|---|---|---|---|
| | | | Phi-3-mini-128K on ∞Bench | | | | | | | |
| FullAttn | 98.64 | 97.12 | 17.92 | 11.16 | 55.46 | 14.83 | 8.00 | 23.10 | 17.43 | 38.18 |
| INFLLM | **100.00** | 97.12 | 14.35 | 4.97 | 38.86 | 11.04 | 3.50 | 25.38 | 15.14 | 34.48 |
| HF-2BITS | 0.00 | 0.00 | 13.80 | 1.44 | 1.75 | 0.20 | 0.50 | 0.00 | 0.57 | 2.03 |
| SIRLLM | 3.39 | 3.39 | **21.06** | 6.32 | 44.98 | **11.99** | 5.00 | 22.34 | 21.71 | 15.58 |
| MINFERENCE | 99.32 | 95.93 | 14.44 | **8.11** | 40.61 | 10.60 | 9.00 | 15.48 | 15.43 | 32.25 |
| **LOCRET** | **100.00** | 97.46 | 16.82 | 7.61 | 46.29 | 11.31 | **10.00** | 27.92 | 29.71 | **34.73** |
| | | | Llama-3.1-8B-instruct on ∞Bench | | | | | | | |
| FullAttn | 100.00 | 99.32 | 26.79 | 15.06 | 68.12 | 13.40 | 17.00 | 20.56 | 34.00 | 43.81 |
| INFLLM | **100.00** | **100.00** | 24.24 | 14.21 | 51.97 | 10.76 | 11.00 | 26.25 | **35.71** | 41.57 |
| HF-2BITS | 36.78 | 6.95 | 8.77 | 4.05 | 27.95 | 3.09 | 5.50 | 13.20 | 22.00 | 14.25 |
| SIRLLM | 1.69 | 1.69 | 25.60 | 8.95 | 55.46 | 10.38 | 9.50 | 23.10 | 3.71 | 15.56 |
| MINFERENCE | **100.00** | 98.47 | 20.64 | 14.35 | 59.83 | **12.20** | 20.50 | 25.89 | 35.43 | 43.03 |
| **LOCRET** | **100.00** | 99.49 | **27.28** | **20.90** | 58.82 | 11.85 | 13.00 | 27.16 | 32.86 | **43.48** |

| Method | CodeU | NQ | CUAD | NarrativeQA | QMSum | SPACE | Avg.↑ |
|---|---|---|---|---|---|---|---|
| | | Phi-3-mini-128K on L-Eval | | | | | |
| FullAttn | 8.89 | 59.14 | 30.34 | 17.59 | 16.05 | 14.51 | 24.42 |
| INFLLM | 5.56 | 34.32 | 14.53 | 14.80 | 13.31 | 14.81 | 16.22 |
| HF-2BITS | 0.00 | 1.69 | 6.40 | 2.04 | 2.73 | 3.34 | 2.70 |
| SIRLLM | **8.89** | 37.92 | 20.89 | 14.51 | 13.70 | 14.46 | 18.40 |
| MINFERENCE | 7.78 | 25.21 | **26.64** | 15.14 | **15.78** | **14.87** | 17.57 |
| **LOCRET** | **8.89** | 51.49 | 22.23 | **16.42** | 14.86 | 14.06 | **21.33** |
| | | Llama-3.1-8B-instruct on L-Eval | | | | | |
| FullAttn | 10.0 | 66.84 | 38.91 | 23.11 | 18.76 | 16.86 | 29.08 |
| INFLLM | 6.67 | 54.77 | 33.76 | 20.35 | 17.62 | 16.73 | 24.98 |
| HF-2BITS | 1.11 | 29.79 | 18.98 | 9.46 | 14.02 | 13.73 | 14.52 |
| SIRLLM | 5.56 | 58.00 | 35.41 | 21.21 | 17.32 | 16.44 | 25.66 |
| MINFERENCE | 7.78 | 31.80 | 36.93 | 19.44 | 18.14 | 16.76 | 21.81 |
| **LOCRET** | **8.89** | **63.03** | **37.21** | **23.59** | **18.17** | **16.87** | **27.96** |

input and output sequences divided by the processing time, along with the accuracy of the measured task. Since the original settings of some algorithms might lead to Out Of Memory (OOM) errors, we remove some tokens from the middle of the input sequence in those cases, marking these settings with *, and report the valid context length in such scenario. For settings without *, we maximize the chunk size for higher speed when the method utilizes the chunked prefill pattern.

R.PassKey is a task where the model retrieves a 5-digit number from a large amount of irrelevant text, a task we believe to be relatively simple for humans. Thus, we consider the task to have failed if the accuracy falls below 95%. As shown in Table 4, aside from the settings that fail on this task, LOCRET achieves the highest inference speed among all methods that can correctly process R.PassKey. Due to its MHA architecture, `Phi-3-mini-128K` has a larger KV cache, which leads to failures for both HF-2BITS and MINFERENCE. Storing the full KV cache on a single 4090 GPU is infeasible, as it requires 48GB of memory. Although the quantized KV cache is reduced to 6GB, the converting processes between representations requires significant GPU memory for intermediate states, resulting in the failure of HF-2BITS. While INFLLM can run in memory-limited scenarios, its offloading process slows down inference, with I/O becoming the bottleneck in attention calculation. SIRLLM fails due to its inaccurate eviction policy, which cannot correctly identify the 5-digit number. In the GQA model (`Llama-3.1-8B-instruct`), which has a smaller KV cache, the quantized cache can fit within the GPU memory. However, the quantization and dequantization processes become the bottleneck, leading to significantly slower speeds. The performance of INFLLM, SIRLLM, and MINFERENCE is similar to that seen with `Phi-3-mini-128K`. Although MIN-FERENCE benefits from faster encoding speeds, it fails on this task because it cannot process the entire input sequence at once. LOCRET strikes a balance between inference speed and performance, making it a far more suitable solution for long-context scenarios on consumer-grade devices.

## 4.4 HYPERPARAMETER ANALYSIS

To address **Q3**, we examine three key hyperparameters: budget, stabilizer length, and chunk size.

Table 4: Executing R.PassKey on an Nvidia 4090. "tok/s" represents the inference speed, "C.Len" stands for the context length after truncation, and "Acc." represents task accuracy. The highest score among 128K context is marked in **bold**.

| Method | | FullAttn | INFLLM | HF-2BITS | SIRLLM | MINFERENCE | **LOCRET** | HF-2BITS* | MINFERENCE* |
|---|---|---|---|---|---|---|---|---|---|
| Phi-3-mini-128K | tok/s↑ | - | 2276.38 | - | 2352.20 | - | **5080.85** | 1098.51 | 4099.92 |
| | C.Len.↑ | 128K | 128K | 128K | 128K | 128K | **128K** | 30K | 14K |
| | Acc.↑ | OOM | 99.83 | OOM | 1.69 | OOM | **100.00** | 0.00 | 13.56 |
| Llama-3.1-8B-instruct | tok/s↑ | - | 2287.66 | 1365.51 | 1589.75 | - | **3209.10** | 3680.06 | 5135.74 |
| | C.Len.↑ | 128K | 128K | 128K | 128K | 128K | **128K** | 30K | 25K |
| | Acc.↑ | OOM | 100.00 | 35.59 | 1.69 | OOM | **100.00** | 26.78 | 20.34 |

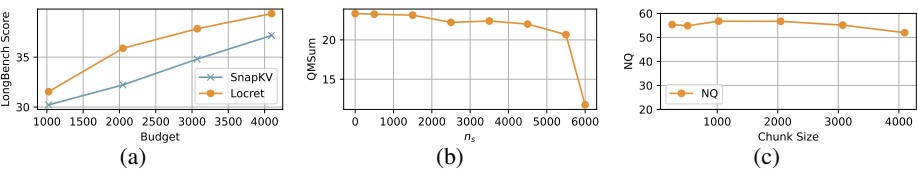

Figure 6: Scores of LOCRET under (a) various budgets; (b) various $n_s$; (c) various chunk size.

**Budget.** To evaluate the robustness of LOCRET under different budget constraints, we compare the proposed method with SNAPKV (Li et al., 2024b) using chunked prefill on LongBench (Bai et al., 2024b). As shown in Figure 6a, when the budget size increases, LOCRET demonstrates a faster performance improvement compared to SNAPKV.

**Stabilizers Length.** As discussed in Figure 4, stabilizers play a crucial role in context retrieval tasks. However, in NLU tasks, the stability of $n_s$ remains relatively high. We evaluate the QMSum dataset from LongBench with different stabilizer lengths $n_s$, with the budget set at 6000. As illustrated in Figure 6b, performance remains consistent when $n_s$ is small. The observed performance degradation at larger $n_s$ values is due to the reduced space available for other cache units.

**Chunk Size.** Executing long-context inference on hardware with varying GPU memory limitations necessitates different chunk sizes. When the chunk size changes, LOCRET demonstrates stable performance. We conduct experiments on the NQ dataset from L-Eval using multiple chunk sizes ranging from 256 to 4096. The results, shown in Figure 6c, highlight the stability of $n_s$.

### 4.5 ORTHOGONALITY TO OTHER METHODS

We evaluate the combination of LOCRET with quantization, token merging and head-wise budget allocation to further enhance LOCRET's efficiency. The experiments demonstrate the compatibility of LOCRET with the aforementioned methods. Further details can be found in Appendix E.

## 5 CONCLUSION & LIMITATION

We propose LOCRET, a lightweight training-based method that enables memory-efficient inference of long contexts on consumer-grade devices. LOCRET introduces retaining heads to predict the CIS of each cache unit during chunked prefill and performs cache eviction based on the predicted CIS. We conduct extensive experiments across different models and multiple datasets to compare LOCRET with major efficient inference techniques, and results show that LOCRET outperforms all baselines, using less GPU memory and without requiring offloading to CPU memory. The framework of LOCRET, including both training and inference, highlights its suitability for low-resource computing scenarios. LOCRET can be applied to various application scenarios, such as end-side multi-modal model inference and context compression during disaggregated inference. In this paper, we explore LOCRET based on two models: `Phi-3-mini-128K` and `Llama-3.1-8B-instruct`, within MHA and GQA architectures, respectively. Future work will involve testing LOCRET on other model architectures, such as encoder-decoder models and multi-latent models. Currently, LOCRET has been evaluated on only two hardware platforms (A800/H800 and 4090), and we plan to extend performance evaluations to other popular hardwares. Additionally, we observe that when the cache budget is extremely limited, LOCRET can degrade to the StreamingLLM pattern (Figure 7). In future work, we will investigate enhancement methods for such scenarios. Additionally, we are interested in integrating LOCRET with other efficient methods, such as offloading and speculative decoding. We also plan to explore how to combine existing query-aware algorithms with LOCRET to achieve more accurate eviction of local tokens.

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

## A  HYPERPARAMETERS, ENVIRONMENT AND BASELINES

### A.1  TRAINING

During the training stage, we first insert retaining head $\mathbf{R}$s to each layar. A retaining head is a small FFN consist of two linear transformations, and the non-linear function is aligned with other non-linears of the conresponding model, with an intermediate size of 1024. We train the appended retaining head $\mathbf{R}$s on the LongAlpaca for 3000 steps with batch size set to 1 and maximum sequence length set to 10240. We use the AdamW scheduler (Loshchilov, 2019) and the learning rate is set to 5e-4. We conduct the training with a linear learning rate scheduler, whose warmup step number is set to 2000. The balance factor between two training loss $\alpha$ is set to 0.0025.

### A.2  INFERENCE

The inference hyperparameters of LOCRET is listed in Table 2. Here, we follow the notations in Algorithm 1. $b$ stands for the cache budget, $B$ is the chunk size of chunked prefill, $n_s$ is the length of stabilizers, and $n_{loc}$ represents the length of locally retained tokens at the end of the input sequence.

Hyperparameters of other baselines are as follows. For INFLLM, we use the recommended settings for Llama-3 to evaluate Llama-3.1. Since there is no recommendations of `Phi-3-mini-128K`, we use the settings for MiniCPM, whose architechture and size is similar to `Phi-3-mini-128K`, to conduct all the experiments. For Quantization, we use the official implementation (Quanto backend) of Hugging Face. For SIRLLM, we set the start size to 4, recent size to 1000 for both models. We set the token entropy size to 6000 and 16384 for `Phi-3-mini-128K` and `Llama-3.1-8B-instruct` respectively. The chunk size of chunked prefill is also 3072 and 1024 for the corresponding model. For MINFERENCE, we utilize the recommended settings for both models.

### A.3  SYSTEM ENVIRONMENT

For all the experiments except the 4090 experiments in Section 4.3, we use a workstation with 8×Nvidia A800/H800 GPUs and 104 Intel(R) Xeon(R) Platinum 8470 CPUs. We only use 1 GPU from the cluster for training, as the GPU requirements are less than 80GB for all training procedures. The device has 1.0 TB CPU memory. The operating system is Red Hat 4.8.5. We conduct all experiments except the full attention full KV cache inference on a single GPU, and 2 GPUs for full attention settings.

For Section 4.3, we conduct the experiments on a single Nvidia 4090 GPU. The device has 512 AMD EPYC 9754 128-Core Processors and 1.0 TB CPU memory. GPUs and CPUs are connected through PCIe Gen 4, which has 16GT/s transmission speed. The operating system is Ubuntu 9.4.0.

### A.4  BASELINES

We compare LOCRET with full attention inference, INFLLM, Quantization, SIRLLM and MINFERENCE. FullAttn inference is performed using vllm (Kwon et al., 2023), which includes automatic tensor parallelism. INFLLM is a representative of the offloading-based methods, where the full KV cache is offloaded to CPU, and the most relavant blocks are retrieved to GPU during inference. For quantization method, we use the Hugging Face implementation of 2-bits KV cache quantization, which is inspired by Liu et al. (2024b), where quantization is conducted along channels instead of tokens. We denote this method as HF-2BITS. SIRLLM is an eviction-based token dropping algorithm, where tokens with low token-entropy is evicted once the cache is fullfilled. We use the official implementation of SirLLM, which includes some CPU operations including importance sorting. MINFERENCE is a typical method of reducing peak GPU memory consumption through rule-based sparse attention, but it does not reduce the size of KV cache. Note that INFLLM, HF-2BITS and SIRLLM does not have official implementation on `Phi-3-mini-128K`, thus we implement these three methods according to the original algorithm. We only use the short factor of RoPE for IN-FLLM, and no further model modification is conducted for HF-2BITS and SIRLLM.

## B  THE GLOBAL AND LOCAL DISCREPANCY OF SCORING FUNCTIONS

Cache importance scoring functions can generally be categorized into two types: causal and non-causal. Non-causal functions, e.g. $H_2O$ and SNAPKV, require information from subsequent cache units to determine the importance score of a cache unit, making them dependent on prefilling the entire sequence. On the other hand, causal functions, e.g. SIRLLM and LOCRET, predict cache importance without relying on subsequent information. Non-causal scoring functions are incompatible with chunked prefill because they cannot calculate scores without access to the full sequence. If such functions are integrated with chunked prefill, they often face a significant discrepancy between the local importance score (without considering subsequent information) and the global importance score (with full context).

To investigate this discrepancy, we measure the consistency of the top 10% most important cache positions identified in prefixes of various lengths compared to the full context. For reference, the full context is truncated to 6K tokens. The results shown in Figure 1 highlights that scoring functions requiring future information, such as $H_2O$ and SNAPKV, suffer from significant discrepancies when subsequent cache units are not considered. SIRLLM, while also causal, shows notable inaccuracies, leading to performance degradation as demonstrated in Table 3 and Table 4.

We also evaluate the end-to-end performance using $H_2O$ and SNAPKV with chunked prefill on $\infty$Bench, shown in Table 5. The results demonstrate that discrepancies between local and global importance scores in H2O and SNAPKV lead to severe performance drops, particularly in R.Number. It is this discrepancy that leads to the failure of H2O and SNAPKV in accurately retrieving information from the context. Specifically, the model is unable to identify the importance of certain cache units at the time they are first encountered. LOCRET, however, avoids such inconsistencies and achieves superior performance.

Table 5: $\infty$Bench scores of $H_2O$, SNAPKV and LOCRET.

| Phi-3-mini-128K on $\infty$Bench | | | | | |
|---|---|---|---|---|---|
| Method | R.Number | E.Sum | E.MC | C.Debug | Avg.↑ |
| FullAttn | 97.12 | 17.92 | 55.46 | 23.10 | 48.40 |
| $H_2O$ | 3.39 | 15.35 | 45.41 | 20.57 | 21.18 |
| SNAPKV | 2.54 | 15.44 | 41.92 | 21.43 | 20.33 |
| LOCRET | **97.46** | **16.82** | **46.29** | **29.71** | **47.57** |

## C  THE EFFECT OF TRAINING

Table 6: The results of LOCRET compared with randomly initialized retaining head $\mathbf{R}$s on $\infty$Bench and L-Eval.

| Phi-3-mini-128K on $\infty$Bench | | | | | | | | | |
|---|---|---|---|---|---|---|---|---|---|
| Method | R.PassKey | R.Number | E.Sum | E.QA | E.MC | Z.QA | E.Dia | C.Debug | M.Find | Avg. |
| Random | 0.00 | 34.00 | 5.09 | 2.68 | 18.34 | 1.54 | 0.00 | 13.71 | 2.57 | 4.92 |
| LOCRET | **100.00** | **97.46** | **16.82** | **7.61** | **46.29** | **11.31** | **10.00** | **27.92** | **29.71** | **34.73** |

We compare the trained LOCRET to appending randomly initialized retaining head $\mathbf{R}$s on $\infty$Bench. The results in Table 6 show that LOCRET training is effective. Randomly initialized of retaining heads give random predictions and evict arbitary cache units at each step, resulting the failure on all tasks.

## D  EVALUATION ON LONGBENCH

We conduct additional experiments to evaluate Locret on LongBench (Bai et al., 2024b), comparing it with baselines such as Full Attention, MInference, InfLLM, and SirLLM. For this evaluation, we used `Phi-3-mini-128K` with a retained head trained on LongAlign. To ensure a fair comparison, we excluded all Chinese subtasks from LongBench and focused solely on the English subtasks, as

`Phi-3-mini-128K` was not specifically trained on Chinese corpora. The results are presented below. For LOCRET , we follow the hyperparameters presented in Table 2.

Table 7: LongBench scores of LOCRET compared with baselines.

| Method | gov_report | triviaqa | narrative qa | qmsum | musique | 2wikimqa | multifield qa_en | repobench -p | qasper | hotpotqa | multi_news | trec | passage_retrieval_en | passage _count | samsum | lcc | Avg.↑ |
|---|---|---|---|---|---|---|---|---|---|---|---|---|---|---|---|---|---|
| FullAttn | 33.35 | 86.38 | 18.21 | 19.51 | 19.82 | 33.37 | 49.82 | 58.02 | 41.07 | 43.06 | **26.57** | 67.00 | 93.50 | 2.97 | 23.15 | 51.86 | 41.73 |
| MINFERENCE | 32.94 | **86.87** | 19.46 | 19.57 | 18.85 | 33.30 | 49.14 | 58.98 | **40.31** | 43.56 | 26.35 | **68.00** | **89.00** | 2.10 | 25.58 | 53.68 | 41.73 |
| SIRLLM | 32.92 | 85.61 | 21.08 | 21.59 | 24.32 | 34.97 | 48.52 | **59.15** | 40.17 | 47.00 | 26.44 | 65.50 | 63.00 | 3.00 | 23.11 | 51.83 | 40.51 |
| INFLLM | 25.96 | 84.87 | 20.83 | 19.61 | 13.63 | 27.43 | 41.29 | 55.73 | 30.51 | 38.05 | 25.36 | 64.50 | 10.00 | **7.50** | 0.28 | **61.59** | 32.95 |
| **LOCRET** | **33.46** | 82.39 | **24.56** | **23.35** | **25.12** | **35.93** | **52.77** | 57.16 | 40.17 | **48.70** | 26.41 | 62.00 | 83.00 | 3.00 | **26.37** | 52.61 | **42.31** |

We also report the maximum memory usage, including the GPU memory, the CPU memory, and the total maximum memory, alongside the average score on LongBench. For FullAttn, we exclude the maximum memory usage, aligning with Figure 5.

Table 8: Comparison of methods on LongBench and memory usage.

| Method | LongBench | Max GPU Memory | Max CPU Memory | Total Max Memory |
|---|---|---|---|---|
| FullAttn | 41.73 | - | - | - |
| MINFERENCE | 41.73 | 27.63 | 0.17 | 27.80 |
| SIRLLM | 40.51 | 18.29 | **0.05** | 18.34 |
| INFLLM | 32.95 | 20.03 | 8.95 | 28.98 |
| **LOCRET** | **42.31** | **17.71** | 0.15 | **17.86** |

From the experiments above, LOCRET demonstrates the best overall performance and excels in the majority of subtasks. It outperforms all the baselines without any noticeable performance degradation while consuming less memory. Although MInference also avoids performance drops, it requires more GPU memory compared to LOCRET. SirLLM achieves comparable memory usage but shows some performance decline compared to FullAttn and LOCRET. InfLLM exhibits the most significant performance drop, and its offloading mechanism results in the highest CPU memory usage, making it the method with the largest total memory consumption. These results highlight LOCRET as an outstanding approach for evaluation on LongBench.

# E   ORTHOGONALITY TO OTHER METHODS

Table 9: Quantization with FullAttn and LOCRET. "M" represents Method and "$-\Delta$" represents the gap of average L-Eval score.

| Setting | M | M-4bits | $-\Delta$ |
|---|---|---|---|
| M=FullAttn | 29.08 | 28.52 | 0.56 |
| M=LOCRET | 27.96 | 27.11 | 0.85 |

Table 10: The average L-Eval scores of LO-COCO, LOCRET, and the combination of LOCOCO and LOCRET.

| Method | LoCoCo | LOCRET | **Combination** |
|---|---|---|---|
| L-Eval | 26.01 | 27.96 | **28.70** |

**KV cache quantization.** According to Zhang et al. (2024d), eviction-based methods like $H_2O$ struggle with compatibility when combined with KV cache quantization. Quantization introduces significant disturbance in the estimation of heavy-hitters, leading to severe performance degradation. However, LOCRET is not affected by such issues and can be combined with quantization while maintaining most of its performance. Here, we compare the performance degradation caused by quantization on LOCRET with that of the full attention method using the same metrics. We use Quanto as the quantization backend and report the average L-Eval score with `Llama-3.1-8B-instruct` as the model backbone. Table 9 shows that the performance drop caused by quantization on LOCRET is only slightly higher than that observed with the full attention method, indicating that LOCRET is a quantization-friendly approach. More details of the experiment are provided in Appendix E.1.

**Token merging.** As described in Section 2, token dropping can also be implemented through an attention pool. Attention pool-based methods (Xiao et al., 2024b; Cai et al., 2024a; Mu et al., 2024; Munkhdalai et al., 2024) merge adjacent tokens or cache units into an attention pool, maintaining a

static cache size. These methods are orthogonal to LOCRET , as the evicted tokens can be merged into a small cache pool and retained in GPU memory. We conduct the following experiment to demonstrate that LOCRET can serve as an effective plug-in scoring function within such frameworks, enhancing performance without increasing memory budget. We select LOCOCO (Cai et al., 2024a) as a representative of the latest attention pool-based methods. LOCOCO maintains a cache set consisting of two parts: the heavy hitters and the convolved non-heavy hitters. During each chunked prefill step, LOCOCO first identifies a set of heavy hitters according to $H_2O$ (Zhang et al., 2024e), then applies 1-D convolution to the non-heavy hitters to compress them into a static size. By replacing $H_2O$'s heavy-hitter scoring function with LOCRET, we retain the cache units with high CIS and convolve the others. We compare this combination with standalone LOCOCO and LOCRET on L-Eval using the `Llama-3.1-8B-instruct` backbone and report the average score across all selected tasks. As shown in Table 10, LOCRET achieves a higher score than LOCOCO, and the combined algorithm outperforms both standalone methods. This suggests that LOCRET provides a more accurate scoring function compared to $H_2O$, and the two methods complement each other, demonstrating their orthogonality. Further details of the experiment are provided in Appendix E.2.

**Head-wise Budget Allocation.** Since LOCRET evict cache units across the attention heads independently, it is compatible with head-wise budget allocation. Here, we combine LOCRET with PYRAMIDKV (Cai et al., 2024b). PYRAMIDKV assumes that identifing the important cache in deeper layers are simpler than shallow layers, thus it allocates more budget to the shallow layers. We evaluate LOCRET+PYRAMIDKV on the following subtasks of ∞Bench using `Phi-3-mini-128K`. Results presented in Figure 11 shows the compatibility of the two methods.

Table 11: ∞Bench scores of the combination of LOCRET and PYRAMIDKV.

| Phi-3-mini-128K on ∞Bench | | | | | |
|---|---|---|---|---|---|
| Method | R.Number | E.Sum | E.MC | C.Debug | Avg.↑ |
| LOCRET | 97.46 | **16.82** | 46.29 | 29.71 | 47.57 |
| LOCRET+PYRAMIDKV | **99.66** | 15.82 | **48.03** | **30.00** | **48.38** |

### E.1 COMBINATION WITH QUANTIZATION

Table 12: L-Eval scores of FullAttn, FullAttn-4bits, LOCRET and LOCRET-4bits. (Detailed)

| Llama-3.1-8B-instruct on L-Eval | | | | | | | |
|---|---|---|---|---|---|---|---|
| Method | CodeU | NQ | CUAD | NarrativeQA | QMSum | SPACE | Avg.↑ |
| FullAttn | 10.0 | 66.84 | 38.91 | 23.11 | 18.76 | 16.86 | 29.08 |
| FullAttn-4bits | 7.78 | 66.64 | 38.25 | 22.76 | 18.85 | 16.84 | 28.52 |
| LOCRET | 8.89 | 63.03 | 37.21 | 23.59 | 18.17 | 16.87 | 27.96 |
| LOCRET-4bits | 4.44 | 63.22 | 36.95 | 22.80 | 18.43 | 16.81 | 27.11 |

We compare the combination of LOCRET and HF-4BITS quantization with the full attention method and the standalong HF-4BITS quantization. We utilize the official implementation of Hugging Face, with Quanto as the backend of quantization. Other hyperparameters are kept same as described in Section 4.1. We conduct the experiment on L-Eval and report the average score, with `Llama-3.1-8B-instruct` backend. The results in Table 12 shows that the degradation caused by quantization is not significantly high, showing that LOCRET exhibits good robustness on data representation and it is friendly to quantization.

### E.2 COMBINATION WITH LOCOCO

Table 13: L-Eval scores of LOCOCO, LOCRET and the combination LOCOCO+LOCRET. (Detailed)

| Llama-3.1-8B-instruct on L-Eval | | | | | | | |
|---|---|---|---|---|---|---|---|
| Method | CodeU | NQ | CUAD | NarrativeQA | QMSum | SPACE | Avg.↑ |
| FullAttn | 10.0 | 66.84 | 38.91 | 23.11 | 18.76 | 16.86 | 29.08 |
| LOCOCO | 4.44 | 61.10 | 35.84 | 19.83 | 18.15 | 16.71 | 26.01 |
| LOCRET | 8.89 | 63.03 | 37.21 | 23.59 | 18.17 | 16.87 | 27.96 |
| LOCOCO+LOCRET | 7.78 | 66.33 | 38.01 | 24.85 | 18.31 | 16.92 | 28.70 |

We compare the combination of LOCOCO and LOCRET with the standalone methods. For LO-COCO, we train the convolution head with the size of convolved cache set to 2048. We extend the context length through chunked prefill training to 64K, which is longer than all tasks' average input length. The convolution kernel is set to 21, and we train the newly-added convolution and layer norms for 200 steps, following the original setting. Since the original `Llama-3.1-8B-instruct` supports 128K context length, we do not modify its positional embedding. During Inference, we keep a cache budget size of 16384. In the standalone LOCOCO setting, there are 2048 cache units are convolved, while the others are the heavy-hitters selected by $H_2O$. In the combined algorithm, we replace $H_2O$ to LOCRET. We select 14336 cache units with the highest CIS, and convolve the other evicted tokens into 2048 cache units. In all methods, we set the local length to 0, following the original setting.

## F  TRAINING ROBUSTNESS

LOCRET demonstrates high robustness to the training settings, suggesting that there is no need for careful tuning of training hyperparameters or meticulous selection of datasets. Here, we ablate the intermediate size of the retaining heads $d_{\mathbf{R}}$ and train the retaining head $\mathbf{R}$s on various long-context tuning datasets to demonstrate the stability of results across different training settings.

### F.1  INTERMEDIATE SIZE OF THE RETAINING HEAD

We align all the training settings as described in Section 4.1 and only change the intermediate size of retaining heads $d_{\mathbf{R}} \in \{256, 512, 1024, 2048, 4096\}$ with the backbone model `Phi-3-mini-128K`. The trained model is evaluated on L-Eval and we report the average L-Eval score corresponding to each intermediate size. Results are listed in Figure 3. The performance variations among all the settings are minimal compared to the changes in the intermediate size, surpassing all baselines in Table 3. This indicates that out method exhibits good performance stability regardless of the intermediate size of the retaining head $\mathbf{R}$s.

Table 14: L-Eval scores with different intermediate size of the retaining head $d_{\mathbf{R}}$. (Detailed)

| $d_{\mathbf{R}}$ | CodeU | NQ | CUAD | NarrativeQA | QMSum | SPACE | Avg.↑ |
|---|---|---|---|---|---|---|---|
| | | | | Phi-3-mini-128K on L-Eval | | | |
| 256 | 8.89 | 51.52 | 23.05 | 16.21 | 15.26 | 13.77 | 21.45 |
| 512 | 6.67 | 50.61 | 23.33 | 16.67 | 15.02 | 14.23 | 21.09 |
| 1024 | 8.89 | 51.49 | 22.23 | 16.42 | 14.86 | 14.06 | 21.33 |
| 2048 | 7.78 | 54.09 | 21.91 | 16.46 | 15.00 | 13.89 | 21.52 |
| 4096 | 10.00 | 52.33 | 23.52 | 16.15 | 14.81 | 14.02 | 21.81 |

We train different retaining head $\mathbf{R}$s with $d_{\mathbf{R}} \in \{256, 512, 1024, 2048, 4096\}$. We keep all the other hyperparameters same, and train on the same dataset. From Table 14, LOCRET shows stability to the intermediate size, in both overall performance and the performance of each single task. While increasing the intermediate size, we observe very slight overall performance enhancement. However, the performance variance is negligible compared to the increase of parameter size, thus we choose to maintain the intermediate size in a small scope to take balance of performance and efficiency.

### F.2  TRAINING DATA INSENSITIVITY

We also consider the sensitivity of the training data, which leads us to ablate the training dataset by training on LongAlign (Bai et al., 2024a) and Anti-Haystack (Pan, 2024), comparing these results with those from LongAlpaca (Chen et al., 2024) in the original training setting. We also align other settings to the original setting and choose the backbone model to be `Phi-3-mini-128K`. We report the average L-Eval score for each training dataset. The results in Table 1 shows that LOCRET has high insensitivity towards different training data. The performance impact of different data recipes is minimal, indicating that our method can be trained on any long-context tuning dataset.

Table 15: L-Eval scores of LOCRET trained on various dataset. (Detailed)

| Phi-3-mini-128K on L-Eval | | | | | | | |
|---|---|---|---|---|---|---|---|
| Dataset | CodeU | NQ | CUAD | NarrativeQA | QMSum | SPACE | Avg.↑ |
| LongAlpaca | 8.89 | 51.49 | 22.23 | 16.42 | 14.86 | 14.06 | 21.33 |
| LongAlign | 10.00 | 55.13 | 21.34 | 16.40 | 15.01 | 14.09 | 22.00 |
| Anti-Haystack | 8.89 | 52.91 | 20.87 | 13.73 | 13.84 | 14.10 | 20.72 |

We conduct training on various datasets and benchmark the trained weights on L-Eval with `Phi-3-mini-128K` backbone, to show the stability towards training datasets. For each datasets, we set the training hyperparameters same and truncate the context to 10240 tokens. We train the first 3000 steps of LongAlpaca and LongAlign. Since Anti-Haystack is a relatively smaller dataset, we utilize the whole dataset, which consist of 2424 entries. The results in Table 15 shows that different training dataset recipe exhibits minor effect towards the overall performance. LOCRET can obtain competitive performance without delicately selecting the training data.

# G EXTREMELY LONG CONTEXT EVALUATION

We create a dataset similar to ∞Bench's R.Number, with an average length of 10 million tokens. Each data point contains a 10-digit number string inserted into an irrelevant context, and the task is to retrieve the inserted number. The dataset consists of 50 examples, with the number strings uniformly distributed throughout the context. We used the hyperparameters from Table 2, with the exception of setting the chunk size to 10240 to speed up inference. The results, presented below in Table 16, show that Locret can efficiently process extremely long contexts. In this experiment, the cache budget is set to 6000, and the compression ratio is 1747.6×.

Table 16: Inference speed with Retaining Heads.

| Phi-3-mini-128K on 10M context | |
|---|---|
| Dataset | R.PassKey_10M |
| LOCRET | 100.00 |

# H COMPRESSING MULTI-TURN CONVERSATIONS

Compared to query-aware eviction methods, such as SNAPKV (Li et al., 2024b), LOCRET is a more suitable solution for multi-turn conversation scenarios. This is because the evaluation of cache importance in LOCRET is based on the cache itself, rather than being dependent on the subsequent query. To demonstrate this, we evaluate LOCRET on the Rock-Paper-Scissors benchmark introduced in SIRLLM (Yao et al., 2024). Since SIRLLM is specifically designed for such scenarios, we use it as our baseline in this benchmark. Results in Table 17 show that Locret is also effective in multi-turn conversation contexts.

The hyperparameters are aligned with those used in SIRLLM, with the cache budget set to 1024, and no stabilizers are retained, as SIRLLM does not retain local tokens in this benchmark. We perform 2000 turns as same as the original SIRLLM settings. The results are presented below.

Table 17: Rock-Paper-Scissors scores of LOCRET and SIRLLM.

| Phi-3-mini-128K on Rock-Paper-Scissors | | | | | | | | | | | |
|---|---|---|---|---|---|---|---|---|---|---|---|
| Preference | Rock | | | Paper | | | Scissors | | | Avg. | |
| | win | tie | lose | win | tie | lose | win | tie | lose | win↑ | lose↓ |
| SIRLLM | 40.00 | 31.75 | 28.25 | 27.5 | 36.55 | 35.96 | 29.35 | 25.15 | 45.50 | 32.28 | 36.57 |
| **LOCRET** | 18.95 | 50.00 | 31.05 | 30.35 | 19.45 | 50.20 | 52.05 | 27.25 | 20.70 | **33.78** | **33.98** |

## I  DISCONTINUOUS CONTEXT AND STABLIZERS

Evicting cache units results in context discontinuity, which causes unstable CIS prediction and inaccurate calculation of later tokens. Thus, we always retain the stabilizers, which are consist of the last $n_s$ cache units in each chunked prefill step. We ablate $n_s$ on R.Number of $\infty$-Bench in the proposed algorithm to demonstrate the necessity of incorporating stabilizers in the design. The results in Figure 4a show that lower stabilizer length $n_s$ causes severe performance degradation and the model fails completely when the stabilizers are absent. We report the maximum absolute error of the last hidden state of the input prompt across different layers in Figure 4b. Large errors can be observed when the stabilizers are short or absent. We also report the mean absolute error of the predicted causal importance values with different stabilizer lengths, compared to the case without evicting any cache units, in Figure 4c. We also observe high errors when the stabilizer length is limited. This explains the reason for failure when the stabilizers are short or absent: context discontinuity leads to instability in the prediction of CIS, resulting in errors during cache eviction and amplifying errors in the hidden states.

## J  RETAINING HEADS DO NOT SLOW DOWN INFERENCE

We evaluate the model's forward throughput under varying context lengths, both with and without retaining heads. The results are summarized below in Table 18. "$\mathbf{R}$" represents the retaining heads, and the throughput is reported in tokens per second (tok/s) in the format "Avg. / Std."

Table 18: Inference speed with Retaining Heads.

| Context Length | 1024 | 2048 | 3072 | 4096 |
|---|---|---|---|---|
| w/o $\mathbf{R}$ Speed | 18674 / 443 | 19743 / 464 | 19982 / 402 | 20304 / 187 |
| w/ $\mathbf{R}$ Speed | 17118 / 1117 | 18503 / 546 | 19054 / 283 | 19153 / 174 |

From the results, no significant latency increase is observed when using retaining heads. The numerical differences are attributed to systematic variations rather than additional overhead introduced by retaining heads during inference.

## K  CAUSAL IMPORTANCE SCORE SIMULATES A CACHE PROBLEM

In this section, we show that assigning each cache unit a CIS and calculate each cache units with top-$b$ cache units simulates a cache problem, i.e. the calculation process can be done in a cache. Thus, LOCRET mathmatically equals to top-$b$ sparse attention.

**Definition K.1.** (Causal Calculation) Given a sequence of objects $c_1, c_2, \cdots, c_n$, if

$$\forall 1 \le i \le n, c_i = f(c_1, c_2, \cdots, c_{i-1})$$

then $f$ is a causal calculation. $c_1, c_2, \cdots, c_n$ is the generated sequence respective to $f$.

For all causal calculations, we can easily split the function into two parts: a selection function and a another function. Formally,

$$\forall \text{ causal calculation f, } \exists \text{ function } g, Sel,$$
$$g : 2^{\{c_1, c_2, \cdots, c_n\}} \to \{c_1, c_2, \cdots, c_n\},$$
$$Sel : 2^{\{c_1, c_2, \cdots, c_n\}} \to 2^{\{c_1, c_2, \cdots, c_n\}}; \ X \mapsto Y \subseteq X,$$
$$\text{s.t. } f = g \circ Sel.$$

**Definition K.2.** (Causal Importance Score) Given a causal calculation $f$ and $c_1, c_2, \cdots, c_n$ is the generated sequence of $f$. $s_1, s_2, \cdots, s_n \in \mathbb{R}$ is a sequence of numbers. If

$$s_i = h(c_i),$$

then $\{s_i\}$ is a CIS of sequence $\{c_i\}$. $h$ is a causal importance scoring function.

**Definition K.3.** (Cache Problem) Given a causal calculation $f = g \circ Sel$, its generated sequence $\{c_i\}$ and a positive number $b \in \mathbb{Z}_+$, if $f$ satisfies the following two condion, then $(f, b, \{c_i\})$ is a cache problem with budget $b$.

- $\forall 1 \leq i \leq n, \ |Sel(c_1, \cdots, c_n)| \leq b$,

- $\forall 1 \leq m_1 < m_2 \leq n, \ Sel(c_1, \cdots, c_{m_2}) \backslash Sel(c_1, \cdots, c_{m_1}) \subseteq \{c_{m_1+1}, \cdots, c_{m_2}\}$.

**Theorem K.1.** (Calculating cache units with Top-$b$ CIS is a cache problem.) Given a causal calculation $f = g \circ Sel$, and its generated sequence $\{c_i\}$, a CIS $s_i = h(c_i)$ and a positive number $b \in \mathbb{Z}_+$, if the selection function $Sel$ satisfies the following condition,

$$Sel(c_1, c_2, \cdots, c_i) = \{c_{p_1}, c_{p_2} \cdots, c_{p_{b'}}\}, \ s_{p_1}, s_{p_2} \cdots, s_{p_{b'}} \in \text{Top-}b(s_1, s_2, \cdots, s_i)$$

then $(f, b, \{c_i\})$ is a cache problem with budget $b$.

**Proof.** (1) For all $i$ of $1 \leq i \leq n$, $|Sel(c_1, \cdots, c_i)| = |\{c_{p_1} \cdots, c_{p_{b'}}\}| = |\{s_{p_1}, \cdots, s_{p_{b'}}\}|$. Since $s_{p_1}, s_{p_2} \cdots, s_{p_{b'}} \in \text{Top-}b(s_1, s_2, \cdots, s_i)$, $|\{s_{p_1}, \cdots, s_{p_{b'}}\}| \leq b$. Thus $|Sel(c_1, \cdots, c_i)| \leq b$.

(2) For all $1 \leq m_1, < m_2, \leq n$,

$$Sel(c_1, \cdots, c_{m_2}) \backslash Sel(c_1, \cdots, c_{m_1}) \subseteq \{c_{m_1+1}, \cdots, c_{m_2}\}$$
$$\Longleftrightarrow \{s_{p_1}, \cdots, s_{p_{m_2}}\} \backslash \{s_{q_1}, \cdots, s_{q_{m_1}}\} \subseteq \{s_{m_1+1}, \cdots, s_{m_2}\}.$$

Assume $\exists s \in \{s_{p_1}, \cdots, s_{p_{m_2}}\} \backslash \{s_{q_1}, \cdots, s_{q_{m_1}}\}$ but $s \notin \{s_{m_1+1}, \cdots, s_{m_2}\}$. Since $s_{p_1}, \cdots, s_{p_{m_2}} = \text{Top-}b(s_1, \cdots, s_{m_2})$, $s \in \{s_1, \cdots, s_{m_2}\}$. Thus $s \in \{s_1, \cdots, s_{m_1}\}$. $s$ is not in the Top-$b$ values of first $m_1$ scores, thus there exists $b$ values larger than $s$, denote as $s_{l_1}, \cdots, s_{l_b}$. Then, $s_{p_1}, \cdots, s_{p_{m_2}} = \text{Top-}b(s_{l_1}, \cdots, s_{l_b}, s_{m_1+1}, \cdots, s_{m_2})$. From this, we can obtain that $\min\{s_{p_{m_2}}\} \geq \min\{s_{l_1}, \cdots, s_{l_b}\} > s, s \notin \{s_{p_1}, \cdots, s_{p_{m_2}}\}$. Contradiction. Finally, there must be $s \in \{s_{m_1+1}, \cdots, s_{m_2}\}$. From (1)(2), $f$ satisfies the two conditions of cache problem. Thus, calculating cache units with Top-$b$ CIS is a cache problem.

## L   RETAINED PATTERNS OF LOCRET

We investigate the retained patterns of LOCRET. We trace the cache units at each attention head through the chunked prefill on R.Number, M.find and E.MC of $\infty$Bench with backbone `Phi-3-mini-128K`, and investigate the pattern variation among different layers on R.Number. We display the results in Figure 7 and Figure 8. The yellow parts are the retained cache, where the y-axis represents cache position and x-axis is the time axis.

Figure 7 shows that the pattern is mostly decided by the tasks, where both heads shows similar pattern in the same task. In R.Number, we are able to observe a strong signal between token 10000 and 15000, which is the position of the inserted number string, indicating that LOCRET can identify the potentially answer-related parts by giving high predicted values of CIS. In M.Find, we can observe the StreamingLLM (Xiao et al., 2024b) pattern, where the tokens at the beginning of the sequence are always important. This is also mentioned as the $\Lambda$-pattern in MINFERENCE. We can also discover the vertical lines in the middle of the sequence. This pattern is also approached by MINFERENCE (Jiang et al., 2024a) by the pattern "vertical-and-slash". In E.MC, $H_2O$ (Zhang et al., 2024e) and ScissorHands (Liu et al., 2024a) pattern can be observed, following the assumption that if a token is activated at some point, it will continue to be activated in the consequencing process. Noticing that the vertical lines always come in groups, which is the fundament of INFLLM (Xiao et al., 2024a) retrieving blocks to calculate. The comparison between two heads also shows that different heads exhibits different features. Head 22 of layer 11 shows stronger vertical lines at some point, where retained pattern of head 14 layer 11 is more even. Head 14 of layer 11 also gives stronger signal to the initial tokens, where this effect is less strong in head 22 layer 11. We also conduct experiments to investigate the patterns across layers. In Figure 8, we show that the pattern variance of the same head in different layers can be large. In shallow layers, e.g. layer 1 and 5, the retained cache units appears to be periodical and semantic independent. However, in middle layers, e.g. layer 13 and 17, the position of the inserted number string is strongly highlighted, indicating that semantic takes over to be the dominant factor. In the deepest layers, e.g. 21, 25 and 29, the highlighted vertical line at the position of the inserted string becomes more accurate.

The retained pattern at different layers shows various features, which might be a good handle to investigate how LLMs understand and process natural language queries.

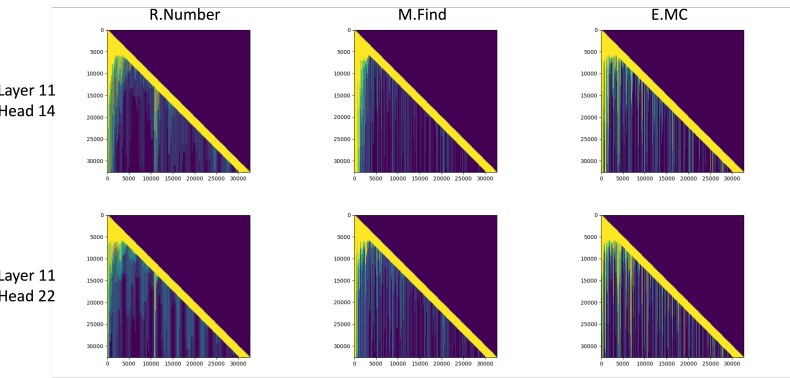

Figure 7: Head patterns across multiple tasks.

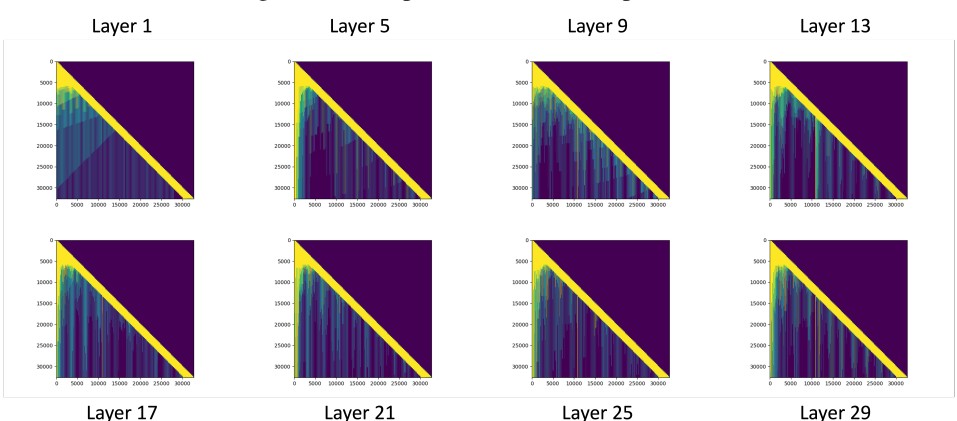

Figure 8: Layer patterns of R.Number

## M    THE TRAINING PROCESS OF LOCRET

Here, we present changing trend of loss and accuracy during training in Figure 9.

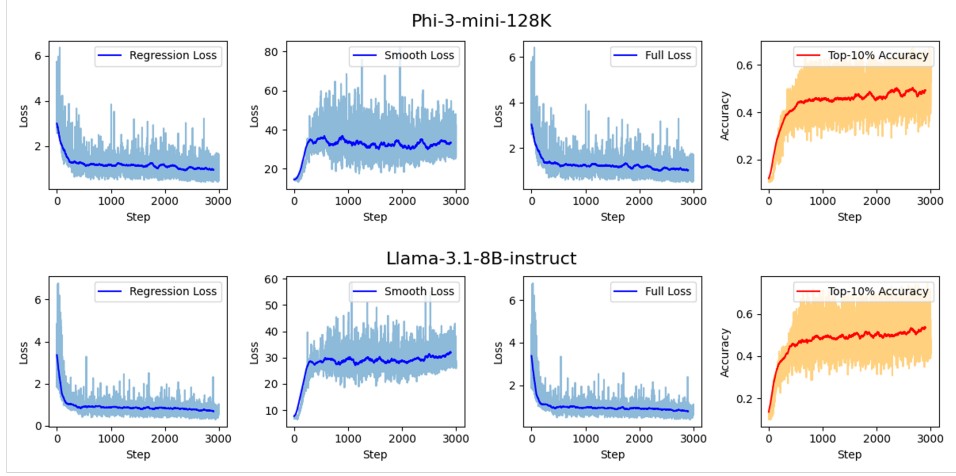

Figure 9: Training loss and accuracy during the training process.

