# OpenReview forum: "Locret: Enhancing Eviction in Long-Context LLM Inference with Trained Retaining Heads"
_ICLR.cc/2025/Conference — Submitted to ICLR 2025_

### Official Review · Reviewer_WKXD · 2024-10-25

**Soundness:** 3
**Presentation:** 2
**Contribution:** 2
**Rating:** 8
**Confidence:** 4

**Summary:**

This paper presents LOCRET, a framework that enhances long-context large language model (LLM) inference on consumer GPUs. LOCRET introduces "retaining heads," lightweight components added to a frozen LLM backbone to estimate the importance of each key-value (KV) cache unit. LOCRET optimizes cache eviction and reduces GPU memory usage during inference by predicting which cache units are crucial. Combined with chunked prefill, it outperforms methods like InfLLM and SirLLM in memory efficiency and generation quality, enabling models like Llama-3.1-8B to run 128K context inference on a single Nvidia 4090 GPU without performance loss.

**Strengths:**

Novelty: The introduction of retaining heads for estimating causal importance is a novel approach to KV cache management.
Practical Impact: Enables deployment of large LLMs on consumer-grade GPUs without significant performance loss.
Comprehensive Evaluation: Extensive experiments across multiple datasets and models validate the effectiveness of LOCRET.
Compatibility: LOCRET can be integrated with other efficient inference methods like quantization and token merging.
Lightweight Training: Requires minimal additional training time and resources

**Weaknesses:**

Clarity of Presentation: The paper contains grammatical errors and unclear notations, hindering understanding.
Theoretical Depth: The theoretical underpinnings, particularly regarding the causal importance score and its properties, could be more thoroughly developed.
Hyperparameter Analysis: Limited discussion on the impact of key hyperparameters (e.g., cache budget, chunk size) on performance.
Limited Discussion of Limitations: The paper does not sufficiently explore potential drawbacks or scenarios where LOCRET may underperform.
Reproducibility: Some essential details for reproducing results are located in the appendix rather than the main text.

**Questions:**

1. Stabilizer Length: Could the authors provide more insight into how the stabilizer length ns affects performance across different models and datasets? Is there an optimal range for ns?
2. Theoretical Justification: Can the authors elaborate on the causal importance score's theoretical properties and explain how it ensures minimal approximation error during cache eviction?
3. Hyperparameter Sensitivity: Have the authors conducted ablation studies on the cache budget b and chunk size B? How do these parameters impact performance and memory usage?
4. Generalization: It might be out-of-scope, but how well does LOCRET generalize to other transformer architectures, such as encoder-decoder models or those with different attention mechanisms?
5. Limitations: Are there specific tasks or contexts where attention pool-based methods might outperform LOCRET? How does LOCRET handle scenarios with severe context discontinuity?
6. Quantization Methods: You mention KV cache quantization techniques, mentioning the computation overhead as their limitation. Could you compare these techniques, e.g., KVQuant, with sparse attention methods such as FastGen?
7. Combination: You mention the possibility of combining your approach with other efficient inference methods. Could you expand on this with results?

---

> ### Author Response · Authors · 2024-11-15
> **Response to Reviewer WKXD (Part 1/3)**
>
> We sincerely appreciate your detailed review and valuable feedback.
>
> ---
>
> ### **W1: Clarity of Presentation: The paper contains grammatical errors and unclear notations, hindering understanding.**
>
> Thank you for your comments. We will conduct a thorough grammar review and ensure that all notations are clearly defined before use in our next revision soon. We are committed to improving the overall quality of our writing.
>
> ---
>
> ### **W2: Theoretical Depth: The theoretical underpinnings, particularly regarding the causal importance score and its properties, could be more thoroughly developed.**
>
> Thank you for pointing this out. I would like to provide the motivation behind designing the CIS, which may help to intuitively understand our proposed methods.
>
> The goal of this paper is to enable long-context inference on consumer-grade devices, particularly in memory-constrained scenarios where GPU memory usage must be strictly controlled. To manage GPU memory consumption, we employ a chunked prefill pattern, which is crucial for reducing memory requirements. However, existing cache importance scoring functions cannot be seamlessly integrated with chunked prefill, as their importance estimation for a token (or cache) depends on subsequent tokens. Locret addresses this limitation by using a causal importance score, meaning the importance score of a cache unit does not depend on subsequent cache units.
>
> A key property of CIS is its causality, which ensures compatibility with chunked prefill. To further explore the incompatibility of existing scoring functions with chunked prefill, we conducted an experiment measuring the consistency of the top 10% most important cache positions in prefixes of varying lengths compared to the full context (truncated to 6K for reference). The results are as follows:
>
> | Prefix Length | 0.5K | 1K | 1.5K | 2K | 2.5K | 3K | 3.5K | 4K | 4.5K | 5K | 5.5K | 6K (full) |
> | - | - | - | - | - | - | - | - | - | - | - | - | - |
> | H2O [1] | 75.46 | 80.58 | 83.30 | 85.00 | 87.20 | 89.22 | 90.76 | 92.48 | 94.02 | 95.54 | 97.40 | 100.00 |
> | SnapKV [2] | 37.12 | 31.17 | 29.20 | 32.09 | 29.60 | 32.71 | 28.80 | 29.20 | 31.82 | 34.86 | 41.65 | 100.00 |
> | SirLLM [3] | 100.00 | 100.00 | 100.00 | 100.00 | 100.00 | 100.00 | 100.00 | 100.00 | 100.00 | 100.00 | 100.00 | 100.00 |
> | Locret | 100.00 | 100.00 | 100.00 | 100.00 | 100.00 | 100.00 | 100.00 | 100.00 | 100.00 | 100.00 | 100.00 | 100.00 |
>
> This experiment demonstrates that scoring functions relying on subsequent information, such as H2O and SnapKV, show significant discrepancies in predicted cache importance when future cache units are not considered. SirLLM, while being a causal importance scoring function, suffers from inaccuracy that leads to substantial performance degradation, as shown in Table 2 and Table 3. Locret, however, avoids these discrepancies and delivers strong performance.
>
> ---
>
> ### **W3: Hyperparameter Analysis: Limited discussion on the impact of key hyperparameters (e.g., cache budget, chunk size) on performance.**
>
> Thank you for your comment. We have included an ablation study on the length of stabilizers (Figure 3) and will incorporate additional ablation studies on cache budget and chunk size in our next revision.
>
> Regarding the impact of cache budget, we conducted experiments with different cache budgets on subsets of LongBench [4], using Phi-3-mini-128K with retaining heads trained on LongAlign. The results indicate that a larger cache budget generally leads to improved performance, although certain tasks are more sensitive to cache budget variations than others.
>
> | Cache budget | 1024 | 2048 | 3072 | 4096 |
> | - | - | - | - | - |
> | GovReport | 27.94 | 30.96 | 31.72 | 32.72 |
> | MultifieldQA-En | 31.63 | 40.44 | 46.95 | 46.2 |
> | PassageRetrieval-En | 31.5 | 60.0 | 69.0 | 75.5 |
> | Dureader | 19.09 | 20.65 | 20.82 | 22.52 |
>
>
> For chunk size, we conducted an experiment evaluating Locret trained with Phi-3-mini-128K LongAlign on the L-Eval's Natural Questions (NQ) dataset using different chunk sizes. The results are as follows:
>
> | Chunk Size | 256 | 512 | 1024 | 2048 | 3072 | 4096 |
> | - | - | - | - | - | - | - |
> | NQ | 55.34 | 54.86 | 56.76 | 56.70 | 55.13 | 51.97 |
>
> This experiment demonstrates the hyperparameter stability of chunk size. A smaller chunk size can be adopted for memory-limited scenarios, such as end-side devices, while a larger chunk size can be used to boost inference speed. Adjustments to chunk size have only a minor impact on performance.

---

> ### Author Response · Authors · 2024-11-15
> **Response to Reviewer WKXD (Part 2/3)**
>
> ### **W4: Limited Discussion of Limitations: The paper does not sufficiently explore potential drawbacks or scenarios where LOCRET may underperform.**
>
> Thank you for your advice. In our paper, we evaluated Locret on two LLMs (Phi-3-mini-128K and Llama-3.1-8B-instruct) and two types of hardware platforms (A800/H800 and 4090). Locret is compatible with various scenarios due to minimal modifications to the model architecture and low hardware requirements.
>
> However, we recognize some potential scenario limitations. For example, the eviction action requires GPU support for the "gather" kernel, making Locret incompatible with some NPU platforms that lack this capability. Additionally, Locret is designed for decoder-only transformer architectures and is not suitable for encoder models or models based on other architectures (e.g., RNNs). We will expand our limitations section to include these and other potential drawbacks in our next revision.
>
> ---
>
> ### **W5: Reproducibility: Some essential details for reproducing results are located in the appendix rather than the main text.**
>
> To enhance readability and provide clearer instructions for reproducing our experiments, we will move key hyperparameters and training/evaluation details from the appendix to the main text. Specifically, we will transfer the essential hyperparameters outlined in Appendix A.1 and A.2 to Section 4.1 (Experimental Setup). Additionally, we will include more detailed instructions on training the retaining heads and evaluating the trained model.
>
> ---
>
> ### **Q1: Stabilizer Length: Could the authors provide more insight into how the stabilizer length ns affects performance across different models and datasets? Is there an optimal range for ns?**
>
> There is a tradeoff between stabilizer length and effective cache budget. Stabilizers occupy space in the retained cache, so a larger stabilizer length reduces the space available for other retained cache units, potentially resulting in the loss of important information and leading to performance degradation. Conversely, a shorter stabilizer length can increase the instability of CIS predictions, leading to more errors during eviction.
>
> From our observations, context retrieval tasks require a larger $n_s$​ due to their need for more accurate CIS prediction and eviction, rather than ample space for important caches. In Figure 3(a), when $n_s$​ is small, there is significant performance degradation in retrieving the correct number. On the other hand, natural language understanding tasks, such as summarization, benefit from shorter $n_s$​ values, as maximizing space for important caches is crucial for better performance.
>
> We conducted an additional experiment on the QMSum task from L-Eval with various stabilizer lengths, keeping the cache budget fixed at 6000. The results demonstrate that overly large $n_s$​ values occupy too much space in the cache, causing performance degradation. The results are as follows:
>
> | $n_s$ | 0 | 500 | 1500 | 2500 | 3500 | 4500 | 5500 | 6000 |
> |-|-|-|-|-|-|-|-|-|
> | QMSum | 23.34 | 23.27 | 23.15 | 22.23 | 22.40 | 22.01 | 20.67 | 11.74 |
>
> We recommend using ~2500 as the optimal value for $n_s$, as shown in Table 6. This value strikes a balance across different types of tasks. We will include the above discussion and this recommendation in our next revision.
>
> ---
>
> ### **Q2: Theoretical Justification: Can the authors elaborate on the causal importance score's theoretical properties and explain how it ensures minimal approximation error during cache eviction?**
>
> The intuition behind the design of training-based CIS prediction is explained in W2, and we have added further experiments to support this discussion.
>
> Defining a definitive "golden label" for the KV cache eviction problem is difficult. Estimating the impact of evicting a group of KV cache units is complex, and to our knowledge, no prior work has specifically focused on this topic. In fact, the most commonly used approaches and the baselines included in this study—such as H2O, SnapKV, SirLLM, and InfLLM [5]—rely on heuristics to evaluate eviction impact or estimate importance based on statistical metrics like attention scores. Consistent with these methods, we identify important cache units using higher attention scores to generate training labels, as a higher attention score generally correlates with greater importance in the attention mechanism.

---

> ### Author Response · Authors · 2024-11-15
> **Response to Reviewer WKXD (Part 3/3)**
>
> ### **Q3: Hyperparameter Sensitivity: Have the authors conducted ablation studies on the cache budget b and chunk size B? How do these parameters impact performance and memory usage?**
>
> Please refer to our response to W3, where we have introduced additional experiments to illustrate the hyperparameter sensitivity of cache budget and chunk size.
>
> ---
>
> ### **Q4: Generalization: It might be out-of-scope, but how well does LOCRET generalize to other transformer architectures, such as encoder-decoder models or those with different attention mechanisms?**
>
> Thank you for this valuable question. Locret is compatible with any transformer-based LLM that utilizes a KV cache. While encoder-decoder models can benefit from Locret by reducing the KV cache burden in the decoder, the encoder part, which does not use a KV cache due to non-causal inference, is not compatible with Locret.
>
> Additionally, other attention mechanisms, such as linear attention and multi-latent attention, are compatible with Locret. Linear attention modifies only the calculation of attention while maintaining the KV cache structure, and multi-latent attention features a single-head KV cache-like structure that is also compatible with Locret.
>
> ---
>
> ### **Q5: Limitations: Are there specific tasks or contexts where attention pool-based methods might outperform LOCRET? How does LOCRET handle scenarios with severe context discontinuity?**
>
> When the budget is extremely limited, such as 128 or 256 tokens, attention pool-based methods can outperform cache eviction methods. This is because eviction-based methods, including Locret, degrade to a StreamingLLM pattern by retaining only the initial and local tokens. As illustrated in Figures 5 and 6, Locret also exhibits a StreamingLLM pattern in these scenarios. Since LoCoCo surpasses StreamingLLM, it can also outperform eviction-based methods under such budget constraints.
>
> However, Table 5 demonstrates the compatibility between LoCoCo and Locret. In scenarios with strict budget constraints, LoCoCo can enhance performance, and when the budget is larger, Locret can further boost standalone LoCoCo's performance. Replacing H2O in LoCoCo with Locret is an effective strategy for improving performance across all scenarios.
>
> Locret assigns continuous position information to the evicted, discontinuous cache to mitigate performance issues related to severe context discontinuity. We are currently exploring training techniques for the LLM backbone to better process discontinuous context, which we plan to address in future work.
>
> ---
>
> ### **Q6 & Q7: Quantization Methods: You mention KV cache quantization techniques, mentioning the computation overhead as their limitation. Could you compare these techniques, e.g., KVQuant, with sparse attention methods such as FastGen? Combination: You mention the possibility of combining your approach with other efficient inference methods. Could you expand on this with results?**
>
> Thank you for this advice. Locret is orthogonal to KV cache quantization, making it compatible with KVQuant. Additionally, Locret is orthogonal to FastGen, which employs a mixture of eviction policies across different heads. Locret can be applied as a specific policy to selected heads to achieve a higher compression ratio.
>
> We are also interested in exploring how Locret can be combined with other efficient inference methods, such as LLM backbone quantization and speculative decoding. Existing works have demonstrated the potential of such combinations; for instance, TriForce [6] integrates H2O-based KV cache compression with speculative decoding, achieving improved decoding throughput. Since Locret can function as a cache importance scoring mechanism, it could be utilized in scenarios where H2O is currently applied. These explorations and combinations involving Locret and other efficient inference methods will be part of our future work.
>
> ---
>
> [1] H2O: Heavy-hitter oracle for efficient generative inference of large language models
>
> [2] SnapKV: SnapKV: LLM knows what you are looking for before generation
>
> [3] SirLLM: Streaming infinite retentive LLM
>
> [4] Longbench: A bilingual, multitask benchmark for long context understanding
>
> [5] InfLLM: Training-Free Long-Context Extrapolation for LLMs with an Efficient Context Memory
>
> [6] Triforce: Lossless acceleration of long sequence generation with hierarchical speculative decoding

---

> > ### Comment · Reviewer_WKXD · 2024-11-19
> > **Response to Authors**
> >
> > Thank you for providing the clarification and additional content. Once the updates are made to the manuscript with the additional content and revisions, I will consider increasing the score.

---

> > > ### Author Response · Authors · 2024-11-19
> > > **Appreciation for Reviewer WKXD's Response**
> > >
> > > We are very grateful your detailed review and swift response. We are currently revising the manuscript and will address all the points mentioned above in the next version. Once we have uploaded the revised version, we will notify you to let you know.

---

> ### Author Response · Authors · 2024-11-19
> **Manuscript updates**
>
> Manuscript updates are as follows. **Note: we use the new figure/table numbering here.**
>
> W2: Figure 1 and line 076-084.
>
> W3, Q3: Figure 6(a) and Figure 6(c).
>
> W4, Q5, Q6, Q7: line 530-539.
>
> W5: line 349-350, line 355-356, and Table 2.
>
> Q1: Figure 6(b).

---

> > ### Comment · Reviewer_WKXD · 2024-11-22
> > **Reply**
> >
> > Updated Score based on the updated revisions

---

> > > ### Author Response · Authors · 2024-11-23
> > > **Official Comment by Authors**
> > >
> > > Thank you so much for your insightful review and feedback! We deeply appreciate the time and effort you put into refining our paper. Your positive comments greatly encourage us as we continue working on building efficient long-context inference methods.
> > >
> > > Once again, thank you for your invaluable support!

---

### Official Review · Reviewer_KmJZ · 2024-10-30

**Soundness:** 3
**Presentation:** 2
**Contribution:** 3
**Rating:** 5
**Confidence:** 5

**Summary:**

This paper proposes LOCRET, a novel framework for long-context LLM inference aimed at reducing GPU memory usage through trained retaining heads. Unlike existing static cache eviction methods, LOCRET uses lightweight training to estimate the causal importance of KV units, achieving more accurate cache eviction. The experimental results demonstrate memory efficiency and competitive generation quality with models like Llama-3.1-8B to perform 128K token inference on a single RTX 4090 GPU.

**Strengths:**

The motivation of the paper is well-articulated, and the experiments are thoughtfully designed. Specifically:

- The claims are strongly supported by comprehensive experimental results. The framework addresses the core issue of KV cache growth through the use of retaining heads, with detailed benchmarks comparing LOCRET against several existing methods.
- The selective eviction strategy, guided by the use of CIS, is convincingly motivated. The experiments are well-structured, thoroughly exploring various datasets, models, and baselines, providing strong evidence of LOCRET’s effectiveness.
- The empirical evaluations comprehensively assess memory usage, inference speed, and performance across a diverse set of tasks. The results are consistently underpinned by sound theoretical analysis. Additionally, LOCRET facilitates long-context inference on GPUs like the Nvidia 4090, significantly enhancing the accessibility of advanced LLMs on consumer-grade hardware.

**Weaknesses:**

The core idea of this paper is to develop an effective eviction policy through training retaining heads. However, several weaknesses need to be addressed:

- SirLLM is not an appropriate baseline for evaluating token eviction strategies. SirLLM is designed primarily for multi-turn conversations and is not tested on benchmarks like InfiniteBench or L-Eval. A more suitable baseline for eviction-based methods would be SnapKV [1]. Although chunk prefilling may not align perfectly with SnapKV, the authors could still avoid OOM errors and reduce GPU peak memory usage by employing layer-by-layer token dropping during prefilling.
- The benchmark suite lacks depth, particularly for information retrieval tasks. The retrieval task within InfiniteBench is overly simplistic, comprising repeated sentences that can be trivially discarded. I recommend that the authors incorporate experiments on RULER [2], following the MInference settings, to provide a more meaningful evaluation of retrieval performance.
- Token eviction based methods may struggle in multi-turn conversation scenarios. For example, in key-value retrieval tasks, if the user queries a different key-value pair during a subsequent turn, the model’s accuracy could degrade significantly due to missing context or prematurely evicted tokens.

If the authors address these concerns, I would consider raising my score.

[1] SnapKV: LLM Knows What You are Looking for Before Generation

[2] RULER: What's the Real Context Size of Your Long-Context Language Models?

**Questions:**

- Could you clarify why there is a significant difference in performance between SirLLM and LOCRET in Table 3? If both methods operate under the same KV budget, the latency bottleneck should primarily stem from the attention operation. What factors contribute to LOCRET’s superior performance despite this similarity?
- Why is it necessary to keep the last $n_s$ caches? Could the retaining head detect and manage these recent tokens effectively? Does this indicate that the retaining head’s predictions are not sufficiently accurate for recent tokens, and if so, what improvements could address this limitation?
- How does LOCRET handle noisy datasets, such as conversational data with inconsistent or off-topic turns? Are there cases where retaining incorrect KV pairs causes irreparable errors during generation, and if so, how does the method mitigate such risks?

---

> ### Author Response · Authors · 2024-11-19
> **Response to Reviewer KmJZ (Part 1/3)**
>
> We sincerely appreciate your detailed review and valuable feedback.
>
> ### **W1.1: SirLLM is not an appropriate baseline. A more suitable baseline for eviction-based methods would be SnapKV.**
>
> Thank you for your suggestion. We have conducted additional experiments using SnapKV [1] with chunked prefill, and our results show that Locret outperforms SnapKV in this scenario. The necessity of eviction with chunked prefill is further explained in W1.2.
>
> Here, we evaluate Locret against SnapKV (as well as H2O [2], a well-established scoring function for cache importance) on various subsets of **InfiniteBench**, utilizing Phi-3-mini-128K. The experiment follows the same setup as outlined in Table 6, with the exception of varying the scoring function. The results are summarized below.
>
> | Method | R.Number | E.Sum | E.MC | C.Debug |
> | - | :-: | :-: | :-: | :-: |
> | H2O | 3.39 | 15.35 | 45.41 | 20.57 |
> | SnapKV | 2.54 | 15.44 | 41.92 | 21.43 |
> | **Locret** | **97.46** | **16.82** | **46.29** | **29.71** |
>
>
>
> Additionally, we test Locret and SnapKV in the chunked prefill scenario on **LongBench** with varying cache budgets. We exclude the Chinese subtasks from LongBench, as the model we are using, Phi-3-mini-128K, is not specifically trained on a Chinese corpus. The retaining heads are trained on the SFT dataset, LongAlign, for 3000 steps. Hyperparameters are consistent with those outlined in Table 6 of our paper.
>
> | Cache Budget | 1024 | 2048 | 3072 | 4096 |
> | - | :-: | :-: | :-: | :-: |
> | SnapKV | 30.22 | 32.22 | 34.81 | 37.17 |
> | **Locret** | **31.54** | **35.89** | **37.85** | **39.36** |
>
> From the experiments above, Locret demonstrates superior performance compared to SnapKV. SnapKV experiences significant performance degradation across all tested subtasks, particularly on context retrieval tasks, such as R.Number in InfiniteBench, highlighting the incompatibility of SnapKV with chunked prefill. In contrast, Locret proves to be a more accurate scoring function under these conditions.
>
> ---
>
> ### **W1.2: Although chunk prefilling may not align perfectly with SnapKV, the authors could still avoid OOM errors and reduce GPU peak memory usage by employing layer-by-layer token dropping during prefilling.**
>
> Thank you for your insightful comment. As demonstrated in the additional experiments in W1.1, SnapKV does not align well with chunked prefill. We believe that eviction with chunked prefill is more efficient than layer-by-layer token dropping, as the latter requires storing a full KV cache for each layer at some point. This behavior can still be resource-intensive when processing longer sequences. For instance, when handling context with a length of 10 million tokens, the full cache for a single layer occupies 120GB of GPU memory, which is difficult to accommodate even with a single A100/H100 GPU (with only 80GB of memory). In contrast, Locret is able to process such large contexts while keeping GPU memory usage within reasonable limits.
>
> Additionally, we would like to include the 10M pass key experiment suggested by Reviewer V3dC. We conducted the experiment using the same hyperparameters as in Table 6, with a budget size of only 6000. The maximum GPU memory used in this setup was less than 18GB.
>
> | Task | R.Number_10M |
> | - | - |
> | Acc. | 100.00 |
>
> The experiment demonstrates that Locret can successfully handle pass key retrieval with a context length of 10 million tokens, and it can be conducted on a single GPU. In contrast, SnapKV with layer-by-layer token dropping requires at least 120GB of GPU memory to store the KV cache for a single layer, consuming significantly more hardware resources. Moreover, Locret is capable of processing streaming context input, where the total context length is infinite. Layer-by-layer token dropping, however, is unsuitable for such scenarios.
>
> We will include the additional experimental results from W1.1 and W1.2 in our next revision and will emphasize the discussion of eviction with chunked prefill versus layer-by-layer token dropping, as it is crucial for clarifying our research objectives.

---

> > ### Author Response · Authors · 2024-11-19
> > **Response to Reviewer KmJZ (Part 2/3)**
> >
> > ### **W2: The retrieval task within InfiniteBench is overly simplistic. I recommend that the authors incorporate experiments on RULER.**
> > Thank you for pointing this out. We have evaluated Locret using Phi-3-mini-128K on the retrieval subtasks of RULER [3]. The training dataset is specifically collected for RULER, featuring shorter context retrieval examples. We evaluated on 100 entries for each subtask, and the reference for FullAttn is based on the results provided in the official RULER repository. The inference hyperparameters are consistent with those in Table 6. The results are presented below. (* indicates results collected from the official RULER repository.)
> >
> > | Task | NIAH-Simple-1 | NIAH-Simple-2 | NIAH-Simple-3 | NIAH-MultiKey-1 | NIAH-MultiKey-2 | NIAH-MultiKey-3 | NIAH-MultiValue-1 | NIAH-MultiQuery-1 | Avg. |
> > | - |:-:|:-:|:-:|:-:|:-:|:-:|:-:|:-:|:-:|
> > | FullAttn* | 98.60 | 97.80 | 97.80 | 86.40 | 65.20 | 42.00 | 66.40 | 69.10 | 77.91 |
> > | Locret | 94.00 | 92.00 | 98.00 | 74.00 | 4.08 | 1.00 | 58.75 | 28.00 | 56.23 |
> >
> > We acknowledge that there is significant performance degradation observed in Locret for some subtasks. For simpler NIAH subtasks, Locret is able to maintain most of its performance. However, for MultiKey NIAH, where the input sequence is dominated by abundant key-value pairs, the compressible part of the context is limited. Since Locret is a query-free compression method, it cannot effectively identify and discard irrelevant portions of the input based on query relevance. For MultiValue and MultiQuery NIAH, where irrelevant context is present, Locret manages to remove some of it.
> >
> > We would like to provide a more detailed discussion of the RULER benchmark. The query-related part of the input context is very sparse, and without access to the query, it is difficult to determine which parts of the context are most important. Locret struggles to perform well in such situations because it conducts eviction along with chunked prefill and operates in a query-free manner. We believe that running RULER within an agent system or through function calls would significantly alleviate this issue, compared to using a single forward pass of an LLM. If string matching were incorporated into the pipeline, we anticipate near-perfect performance. A more effective solution could involve the LLM retrieving the query, followed by an exact string match to identify all relevant answer regions, then processed by the LLM to extract the final answer.
> >
> > For KV cache compression methods, we believe that approaches like Locret are capable of removing semantically redundant parts of the input context, even without query awareness. If Locret were combined with query-aware offloading systems [4], where evicted cache units are moved to CPU memory, it would be possible to retrieve them back to the GPU with query awareness. This remains a topic for future work. Additionally, we are interested in developing a compressed cache representation to store evicted units and "uncompress" them when processing the query at the end of the sequence. We would also sincerely welcome any insights on combining query-unaware cache compression methods with query-aware systems to improve performance on more challenging retrieval tasks.
> >
> > ---
> >
> > ### **W3: Token eviction based methods may struggle in multi-turn conversation scenarios.**
> > Thank you for your comments. Compared to query-aware eviction methods, such as SnapKV, Locret is a more suitable solution for multi-turn conversation scenarios. This is because the evaluation of cache importance in Locret is based on the cache itself, rather than being dependent on the subsequent query. To demonstrate this, we use the Rock-Paper-Scissors benchmark introduced in SirLLM [5], showing that Locret is also effective in multi-turn conversation contexts. SirLLM is specifically designed for such scenarios, and thus we use it as our baseline in this benchmark.
> >
> > The hyperparameters are aligned with those used in SirLLM, with the cache budget set to 1024, and no stabilizers are retained, as SirLLM does not retain local tokens in this benchmark. We perform 2000 turns as same as the original SirLLM settings. The results are presented below.
> >
> > | Preference | Rock | Rock | Rock | Paper | Paper | Paper | Scissors | Scissors | Scissors | Avg. | Avg. |
> > | - |:-:|:-:|:-:|:-:|:-:|:-:|:-:|:-:|:-:|:-:|:-:|
> > | | win | tie | lose | win | tie | lose | win | tie | lose | win | lose |
> > | SirLLM | **40.00** | 31.75 | 28.25 | 27.5 | 36.55 | 35.96 | 29.35 | 25.15 | 45.50 | 32.28 | 36.57 |
> > | Locret | 18.95 | 50.00 | 31.05 | **30.35** | 19.45 | 50.20 | **52.05** | 27.25 | 20.70 | **33.78** | **33.98** |
> >
> > The experimental results demonstrate that Locret outperforms SirLLM when using the same cache budget in the multi-turn conversation scenario. Locret is able to make more accurate cache importance estimations and retain more critical cache units, highlighting its compatibility in such scenarios. We have elaborated on this topic in Q3.

---

> > > ### Author Response · Authors · 2024-11-19
> > > **Response to Reviewer KmJZ (Part 3/3)**
> > >
> > > ### **Q1: Could you clarify why there is a significant difference in performance between SirLLM and LOCRET in Table 3?**
> > >
> > > We directly use the official implementation of SirLLM. From our observations, this implementation is not optimized for inference speed. For instance, when selecting the top-k most important tokens, it uses the CPU to sort all the token entropy scores, rather than performing the top-k operation on the GPU. Additionally, the official implementation involves excessive tensor movement and concatenation, which further reduces speed.
> > >
> > > We will clearly highlight this in our next revision and will make efforts to optimize the official implementation for a more accurate comparison.
> > >
> > > ---
> > >
> > > ### **Q2: Why is it necessary to keep the last caches?**
> > >
> > > It is essential to retain the last caches since the predicted CIS is causal. In the lambda-pattern proposed in MInference [6], the current tokens are always retained. However, because CIS is a causal scoring function, the retaining heads cannot determine when the input sequence reaches the end (since it cannot foresee subsequent content), which prevents it from implementing the lambda-pattern. To address this issue, we ensure that the current tokens are retained. Additionally, we show in Figure 3 that the last caches act as stabilizers and help reduce prediction errors in CIS.
> > >
> > > To eliminate this constraint, integrating non-causal scoring functions, such as H2O and SnapKV, to predict local importance could be a potential solution. We will include this discussion in the "Discussion and Limitations" section of our next revision.
> > >
> > > ---
> > >
> > > ### **Q3: How does LOCRET handle noisy datasets, such as conversational data with inconsistent or off-topic turns? Are there cases where retaining incorrect KV pairs causes irreparable errors during generation, and if so, how does the method mitigate such risks?**
> > >
> > > Thank you for providing such valuable insight. This issue can be alleviated by setting a minimum cache budget for each turn during multi-turn conversation processing. For each turn, we evict cache units until the cache reaches the minimum budget, ensuring that no turn is entirely evicted. Even when future turns shift to a different topic, useful information from previous turns can still be retrieved from the compressed cache.
> > >
> > > Furthermore, since CIS is a causal prediction method, eviction is not dependent on the subsequent query. This means that only the redundant parts within the document itself are evicted. Unlike query-aware methods, this eviction does not involve removing information unrelated to the current query, making it effective for handling inconsistent or off-topic turns.
> > >
> > > ---
> > >
> > > [1] SnapKV: SnapKV: LLM knows what you are looking for before generation
> > >
> > > [2] H2O: Heavy-hitter oracle for efficient generative inference of large language models
> > >
> > > [3] RULER: What's the Real Context Size of Your Long-Context Language Models?
> > >
> > > [4] ShadowKV: KV Cache in Shadows for High-Throughput Long-Context LLM Inference
> > >
> > > [5] SirLLM: Streaming infinite retentive LLM
> > >
> > > [6] MInference 1.0: Accelerating Pre-filling for Long-Context LLMs via Dynamic Sparse Attention

---

> ### Author Response · Authors · 2024-11-19
> **Manuscript updates**
>
> Manuscript updates are as follows. **Note: we use the new figure/table numbering here.**
>
> W1: Figure 6(a), Table 5 and Table 16.
>
> W3, Q3: Table 17.
>
> Q1: line 858-859.
>
> Q2: line 530-539.

---

> ### Comment · Reviewer_KmJZ · 2024-11-25
>
> Thank you to the authors for providing additional experiments and detailed explanations. I appreciate your efforts. I have a few further suggestions and comments:
>
> - **Unfair Evaluation on SnapKV:** I recommend that the authors report the performance of SnapKV on InfiniteBench or RULER (<128K) without chunk prefill, as this scenario would not cause out-of-memory issues with a single A100/H100 GPU (with 80GB of memory). Conducting chunk prefill would not be a fair evaluation for SnapKV, as it is not designed for such usage.
> - **10M Pass Key Experiment:** The 10M pass key experiment appears to be of limited significance because the task is overly simplistic—the “haystack” and “needle” are too distinct, making the task easy to hack through training.
> - **Poor Generalization:** The performance of the proposed method on RULER for complex tasks suffers from accuracy drops, even when trained on datasets specifically curated for RULER. This indicates that the method lacks generalizability.
> - **Unsuitable Multi-Turn Task:** The current evaluation dataset for multi-turn tasks does not seem ideal. As shown, StreamingLLM’s performance is not particularly poor (over 30 points as reported in SirLLM’s paper), and your method’s improvement is marginal. Since StreamingLLM is widely regarded as a weak baseline (e.g., its inability to perform needle retrieval), a better evaluation dataset is needed. A simple alternative could be RULER's multi-key task, where the model retrieves different keys in different turns. However, as the results suggest, Locret appears to struggle with such tasks.
> - **Lack of Off-Topic Turns Experiments:** It would be beneficial to include an experiment that evaluates off-topic turns. For instance, in RULER's QA task with multiple documents, you could ask different off-topic QA questions over multiple rounds before asking the main question. This would help assess whether accuracy drops in such settings.
>
> I hope these suggestions provide useful insights to strengthen your paper further. Thank you again for your effort for conducting additional experiments.

---

> > ### Author Response · Authors · 2024-11-25
> > **Response to Reviewer KmJZ's Official Comment**
> >
> > Thank you for your valuable insights and suggestions. We would like to take this opportunity to clarify certain aspects related to chunked prefill in our work.
> >
> > Our primary goal is to enable long-context inference on consumer-grade devices with limited GPU memory. For instance, the Nvidia 4090 has a memory capacity of only 24GB, making peak GPU memory usage a critical factor in determining whether a task can be executed on such devices. Consequently, our focus is on optimizing the model's performance while minimizing its maximum GPU memory usage.
> >
> > When performing 128K inference using Phi-3-mini-128K, the full KV cache requires 48GB of memory, far exceeding the capacity of a single Nvidia 4090, even without accounting for runtime memory and model weights. Even with layer-by-layer prefill, the KV cache for each layer consumes approximately 1.5GB of memory, which remains prohibitively large.
> >
> > Therefore, we believe that chunked prefill combined with cache eviction represents a crucial strategy in this context. Our work seeks to develop an effective eviction method that integrates seamlessly into this framework. By prioritizing maximum GPU memory usage, our approach differs from other cache compression techniques, which typically focus on metrics such as compression ratio or decoding speed at the system level. We aim to provide an initial solution for these scenarios and plan to focus on enhancing the performance of RULER in our future research.
> >
> > Thank you again for your suggestions. We hope this discussion provides greater clarity on the scope of our work.

---

> ### Comment · Reviewer_KmJZ · 2024-11-25
>
> Thank you for your response. I believe that 1.5GB peak memory consumption is acceptable even for the 4090, which is entirely satisfactory.
>
> If you are not inclined to measure the 128K RULER, you can opt for shorter ones, such as the 32K or 64K RULER. Alternatively, you can choose GQA models. The sequence length is not the key point; the key point is that **you need to measure complex tasks like RULER fairly.**
>
> If the proposed method cannot outperform SnapKV, which is a widely used baseline, I think it is not good enough.

---

### Official Review · Reviewer_j7cb · 2024-11-02

**Soundness:** 3
**Presentation:** 3
**Contribution:** 2
**Rating:** 5
**Confidence:** 4

**Summary:**

This paper proposes a training-based KV cache compression framework LOCRET for long-context LLM inference. The framework introduces retaining heads to evaluate the causal importance of KV cache units, allowing for more accurate eviction within a fixed cache size. The proposed framework is evaluated with two LLMs on Nvidia 4090 GPU.

**Strengths:**

1.	This paper proposes a training-based KV cache compression framework LOCRET for selective KV cache eviction for long-context LLM inference. The proposed framework on two LLMs outperforms related methods on two LLMs and two benchmarks.
2.	The paper is easy to follow.

**Weaknesses:**

1.	The paper claimed “LOCRET is also applicable to all transformer-based LLMs and various hardware”. However, the proposed method is only evaluated with two LLMs (Phi-3-mini-128K and Llam-3.1-8B-instruct) and one hardware platform (Nvidia 4090 GPU).
2.	The proposed framework is validated with ∞Bench and L-Eval. How is the performance on other long-context benchmarks, such as longBench, et al. ?

**Questions:**

Please refer to weaknesses.

---

> ### Author Response · Authors · 2024-11-17
> **Response to Reviewer j7cb (Part 1/2)**
>
> We deeply thank you for your thorough review.
>
> ---
>
> ### **W1: The proposed method is only evaluated with two LLMs (Phi-3-mini-128K and Llam-3.1-8B-instruct) and one hardware platform (Nvidia 4090 GPU).**
>
> Thank you for your valuable comments. The proposed method, Locret, has been evaluated on two LLMs with MHA and GQA architectures. Locret is compatible with any decoder-only LLM that utilizes a KV cache, as the modifications to the model architecture are minimal. Decoder-only Transformers with MHA or GQA attention mechanisms are widely adopted in both academia and industry, making them the most popular LLM architectures at present. We believe our experiments demonstrate that Locret performs effectively with such model architectures.
>
> Regarding the hardware platform, we evaluated Locret on A800/H800 GPUs (Table 2) and the 4090 GPU (Table 3). Locret has minimal hardware requirements, as the algorithm is implemented entirely in PyTorch and does not require any CUDA-level modifications. Therefore, we believe it can be executed on any GPU platform while achieving comparable performance.

---

> ### Author Response · Authors · 2024-11-17
> **Response to Reviewer j7cb (Part 2/2)**
>
> ### **W2: How is the performance on other long-context benchmarks, such as longBench, et al. ?**
>
> Thank you for your valuable advice. We conducted additional experiments to evaluate Locret on LongBench [1], comparing it with baselines such as Full Attention, MInference [2], InfLLM [3], and SirLLM [4]. For this evaluation, we used Phi-3-mini-128K with a retained head trained on LongAlign. To ensure a fair comparison, we excluded all Chinese subtasks from LongBench and focused solely on the English subtasks, as Phi-3-mini-128K was not specifically trained on Chinese corpora. The results are presented below.
>
> | Method             | Avg. Score | gov_report | triviaqa | narrativeqa | qmsum | musique | 2wikimqa | multifieldqa_en | repobench-p | qasper | hotpotqa | multi_news | trec  | passage_retrieval_en | passage_count | samsum | lcc   |
> |--------------------|:----------:|:--------:|:-----------:|:-----:|:-------:|:--------:|:---------------:|:-----------:|:------:|:--------:|:----------:|:-----:|:--------------------:|:-------------:|:------:|:-----:|:---------:|
> | FullAttn          | 41.73     | 33.35      | 86.38 | 18.21       | 19.51 | 19.82   | 33.37    | 49.82           | 58.02       | **41.07** | 43.06    | **26.57**  | 67.00 | **93.50**           | 2.97          | 23.15  | 51.86 |
> | **Locret**            |**42.31** | **33.46**  | 82.39    | **24.56**   | **23.35** | **25.12** | **35.93** | **52.77**       | 57.16       | 40.17  | **48.70** | 26.41      | 62.00   | 83.00                  | 3.00      | **26.37**  | 52.61 |
> | MInference        | 41.73     | 32.94      | **86.87** | 19.46       | 19.57 | 18.85   | 33.30    | 49.14           | 58.98   | 40.31  | 43.56    | 26.35      | **68.00** | 89.00                  | 2.10          | 25.58  | 53.68 |
> | SirLLM            | 40.51     |32.92      | 85.61    | 21.08       | 21.59 | 24.32   | 34.97    | 48.52           | **59.15**       | 40.17  | 47.00    | 26.44      | 65.50 | 63.00                  | 3.00          | 23.11  | 51.83 |
> | InfLLM            | 32.95     |25.96      | 84.87    | 20.83       | 19.61 | 13.63   | 27.43    | 41.29           | 55.73       | 30.51  | 38.05    | 25.36      | 64.50 | 10.00                  | **7.50**          | 0.28   | **61.59** |
>
> We also report the maximum memory usage, including GPU memory, CPU memory, and total maximum memory, alongside the average score on LongBench. For FullAttn, we exclude the maximum memory usage, aligning with Figure 4.
>
> | Method | LongBench | Max GPU Memory | Max CPU Memory | Total Max Memory |
> |-|:-:|:-:|:-:|:-:|
> | FullAttn | 41.73 | - | - | - |
> | **Locret** | **42.31** | **17.71** | 0.15 | **17.86** |
> | MInference | 41.73 | 27.63 | 0.17 | 27.80 |
> | SirLLM | 40.51 | 18.29 | **0.05** | 18.34 |
> | InfLLM | 32.95 | 20.03 | 8.95 | 28.98 |
>
> From the experiments above, Locret demonstrates the best overall performance and excels in the majority of subtasks. It outperforms all the baselines without any noticeable performance degradation while consuming less memory. Although MInference also avoids performance drops, it requires more GPU memory compared to Locret. SirLLM achieves comparable memory usage but shows some performance decline compared to FullAttn and Locret. InfLLM exhibits the most significant performance drop, and its offloading mechanism results in the highest CPU memory usage, making it the method with the largest total memory consumption. These results highlight Locret as an outstanding approach for evaluation on LongBench.
>
> We plan to integrate the experiments mentioned above into the experimental section in our next revision. The comparison of LongBench scores and memory consumption, along with the accompanying discussion, will also be included.
>
> ---
>
> Thank you for your valuable insights on improving our paper by incorporating additional experiments. We hope this response thoroughly addresses your questions. If there are any remaining concerns, we are happy to provide further clarifications or conduct additional experiments. We kindly request a raise of the overall rating if all issues have been addressed.
>
> [1] LongBench: A Bilingual, Multitask Benchmark for Long Context Understanding
>
> [2] MInference 1.0: Accelerating Pre-filling for Long-Context LLMs via Dynamic Sparse Attention
>
> [3] SirLLM: Streaming infinite retentive LLM
>
> [4] InfLLM: Training-Free Long-Context Extrapolation for LLMs with an Efficient Context Memory

---

> ### Author Response · Authors · 2024-11-19
> **Manuscript updates**
>
> Manuscript updates are as follows. **Note: we use the new figure/table numbering here.**
>
> W1: line 858-859.
>
> W2: Table 7, Table 8.

---

> ### Author Response · Authors · 2024-11-25
> **Looking forward to futher discussions with Reviewer j7cb**
>
> Dear Reviewer j7cb,
>
> We would like to appreciate our gratefulness again for your valuable reviews. We have already uploaded the revised version based on your suggestions. Since the discussion period will be ending soon, we are looking forward to discussing the newly added content and our response with you. Should our responses have addressed your concerns, we would be grateful for an improved score. Thanks again for your time and effort.
>
> Submission 5591's Authors

---

> ### Author Response · Authors · 2024-12-02
> **Additional Results of Llama-3.2-1B-instruct**
>
> Thanks for you advice on testing more models. We conduct experiments using the LongBench dataset on Llama-3.2-1B-instruct, and compare Locret with InfLLM and SirLLM, following the LongBench experiments with Phi-3-mini-128K above. We skip MInference, as MInference does not officially support Llama-3.2-1B-instruct (they have not provided the official configuration of heads).
>
> The results are presented below. (The highest score among Locret, SirLLM and InfLLM is marked in bold font, and FullAttn is provided as a reference.)
>
> | Method             | Avg. Score | gov_report | triviaqa | narrativeqa | qmsum | musique | 2wikimqa | multifieldqa_en | repobench-p | qasper | hotpotqa | multi_news | trec  | passage_retrieval_en | passage_count | samsum | lcc   |
> |--------------------|:----------:|:--------:|:-----------:|:-----:|:-------:|:--------:|:---------------:|:-----------:|:------:|:--------:|:----------:|:-----:|:--------------------:|:-------------:|:------:|:-----:|:---------:|
> | FullAttn          | 31.92     | 28.64	| 80.35	| 18.94	| 22.11	| 19.57	| 28.16	| 42.29	| 43.64	| 15.33	| 35.82	| 25.37	| 63.50	| 4.45	| 3.50	| 39.99	| 39.09 |
> | **Locret**        | **31.21**         | **28.00** | **81.53**	| **19.24**	| **21.73**	| **18.88**	| **28.94**	| **41.39**	| 42.12	| **15.94**	| **35.56**	| **25.45**	| 57.00	| **2.92**	| **4.00**	| **38.33**	| 38.40 |
> | SirLLM            | 25.24         | 26.63	| 46.16	| 7.06	| 19.45	| 4.41	| 27.89	| 38.95	| 37.61	| 15.62	| 17.58	| 25.37	| **62.00**	| 2.50	| 2.79	| 30.87	| 38.89 |
> | InfLLM            | 29.22         | 23.08	| 74.65	| 15.42	| 20.16	| 11.36	| 26.72	| 32.51	| **51.00**	| 15.51	| 30.36	| 24.20	| 55.50	| 0.50	| 3.50	| 33.90	| **49.15** |
>
> | Method | LongBench | Max GPU Memory | Max CPU Memory | Total Max Memory |
> |-|:-:|:-:|:-:|:-:|
> | FullAttn | 31.92| - | - | - |
> | **Locret** | **31.21** | **6.85** | 0.45 | **7.30** |
> | SirLLM | 25.24 | 10.24 | **0.36** | 10.60 |
> | InfLLM | 29.22 | 11.83 | 0.89 | 12.72 |
>
> From the results above, Locret also shows compatibility on Llama-3.2-1B-instruct. It only exhibits neglegible performance drop, while SirLLM and InfLLM shows large performance degredation. Locret also uses the less overall memory compared with the tested baselines. We will integrate the additional results to the next revision (maybe the camera ready version, as we cannot upload pdf now) of our paper.
>
> We hope such additional results are able to show the generalizability of the proposed algorithm, Locret. It can be applied to the majority of decoder-only LLMs. We are also glad to test more settings if there are further needs.

---

> > ### Author Response · Authors · 2024-12-03
> > **Looking forward to futher discussions with Reviewer j7cb**
> >
> > Dear Reviewer j7cb,
> >
> > We sincerely thank you for you thoroughful review. We kindly remind you that the discussion phase will be ending in 10 minutes. We have provided more experimental results on Llama-3.2-1B-instruct, where the generalizability of Locret on various models is prooved. We are eager to have an extended discussion with you about the newly added content, and it is our pleasure to make futher additional experiments to enhance our work. Thanks again for your time and effort, and we wish you a happy holiday and New Year.
> >
> > Submission 5591 "Locret: Enhancing Eviction in Long-Context LLM Inference with Trained Retaining Heads" 's authors.

---

### Official Review · Reviewer_xs3N · 2024-11-02

**Soundness:** 2
**Presentation:** 3
**Contribution:** 3
**Rating:** 3
**Confidence:** 5

**Summary:**

To address the substantial overhead of KV cache in long-context reasoning with large language models, this paper introduces a novel method named LOCERT for KV cache pruning. LOCERT utilizes a more precise pruning metric called the causal importance score (CIS) to preserve the most significant KV cache entries.

**Strengths:**

- The method proposes a lightweight training-based selective key-value cache eviction paradigm for long-context language model inference, with an offline training cost of less than 1 GPU hour.
- Extensive validation on various datasets confirms the superiority of our proposed method over the baselines discussed in the paper.
- An efficient inference system implementation is provided, integrating a retaining head mechanism into a segmented pre-filling inference framework. It maintains a fixed-size cache set by evicting cache units with low predicted importance, thereby controlling GPU memory usage.
- The paper discusses the inadequacies of existing methods such as KV quantization, which fail to address the overhead caused by linear growth in KV size. Our selection-based KV cache eviction method utilizes a static-sized KV cache and outperforms previous strategies in preserving important KV cache entries.

**Weaknesses:**

- The proposed method requires additional training, and although the authors claim it only needs one hour, it also utilizes an eight-card A800 server, which is still resource-intensive.
- The novelty of the proposed method is modest. It is unclear why the training of heads to perform KV cache eviction, predicting each KV's importance, and using the causal importance score (CIS) for pruning, is superior to existing methods like H2O.
- The paper lacks a detailed analysis of the causal importance score (CIS) and needs a deeper discussion to explain why this metric effectively reflects the importance of KV cache.

**Questions:**

- Regarding the use of a static-sized KV cache in selection-based KV cache eviction methods, can you explain why "the weakening correlation between local and global importance as sequences grow exacerbates this issue"?
- During training, the first loss term merely learns the maximum value of each column in the attention score. How effective would it be to directly use the maximum value of each column as a metric during inference?
- The paper mentions that methods like H2O cannot be effectively combined with KV quantization approaches. What are the actual performances of these methods?
- There are many papers similar to H2O that use attention score statistics for pruning, such as SnapKV and PyramidKV [2]. How does the method proposed in this paper compare with these approaches?
- Is the Stabilizer used only for selecting recent tokens?
- Is the performance improvement in this paper due to the SFT? What would be the effect if SFT were directly applied to the model?
- Should the number of heads in a retaining head be the same as in Query, or should it match Key/Value? If it matches Query, in structures like Grouped-Query Attention where each head's Key/Value corresponds to multiple heads' Query, how did you train this setup?

[1] LLM Knows What You are Looking for Before Generation
[2] Dynamic KV Cache Compression based on Pyramidal Information Funneling

---

> ### Author Response · Authors · 2024-11-15
> **Response to Reviewer xs3N (Part 1/3)**
>
> We sincerely appreciate your detailed review and valuable feedback.
>
> ---
>
> ### **W1: The additional training in the proposed method is still resource-intensive.**
>
> We addressed the cost of additional training for Locret in lines 344–345, where we specify that the training for both models in our benchmark requires **less than 1 GPU hour on a single A800 GPU**.
>
> Although Appendix A.2 (System Environment) mentions that our experiments were conducted on an 8*A800/H800 GPU cluster, we want to clarify that only a single GPU was utilized for training. We acknowledge the potential confusion and appreciate your observation. We will make this clearer in our next revision.
>
> ---
>
> ### **W2: Novelty concern. It is unclear why to train the heads to obtain CIS instead of using existing methods like H2O.**
>
> We appreciate your thoughtful feedback. Below, we explain our rationale for employing a training-based approach for predicting CIS and performing eviction, instead of directly using existing cache importance scoring functions such as H2O.
>
> Our primary objective is to enable long-context inference on consumer-grade devices, particularly under memory-constrained conditions where GPU memory usage must be strictly controlled. To achieve this, we utilize a chunked prefill pattern that is essential for reducing memory consumption. However, existing cache importance scoring methods cannot be adapted to chunked prefill due to their reliance on subsequent tokens for estimating the importance of a token or cache.
>
> Locret addresses this limitation by employing a causal importance scoring mechanism, where the importance score of a cache unit does not depend on future units. This feature allows seamless integration with chunked prefill, solving the issue posed by non-causal methods.
>
> To further illustrate the incompatibility of existing scoring functions with chunked prefill, we conducted an experiment measuring the consistency of the top 10% most important cache positions in prefixes of varying lengths compared to a 6K full context. The results are shown below:
>
> | Prefix Length | 0.5K | 1K | 1.5K | 2K | 2.5K | 3K | 3.5K | 4K | 4.5K | 5K | 5.5K | 6K (full) |
> | - | - | - | - | - | - | - | - | - | - | - | - | - |
> | H2O [1] | 75.46 | 80.58 | 83.30 | 85.00 | 87.20 | 89.22 | 90.76 | 92.48 | 94.02 | 95.54 | 97.40 | 100.00 |
> | SnapKV [2] | 37.12 | 31.17 | 29.20 | 32.09 | 29.60 | 32.71 | 28.80 | 29.20 | 31.82 | 34.86 | 41.65 | 100.00 |
> | SirLLM [3] | 100.00 | 100.00 | 100.00 | 100.00 | 100.00 | 100.00 | 100.00 | 100.00 | 100.00 | 100.00 | 100.00 | 100.00 |
> | Locret | 100.00 | 100.00 | 100.00 | 100.00 | 100.00 | 100.00 | 100.00 | 100.00 | 100.00 | 100.00 | 100.00 | 100.00 |
>
> This experiment highlights that scoring functions requiring future information, such as H2O and SnapKV, suffer from significant discrepancies when subsequent cache units are not considered. On the other hand, SirLLM, while also causal, shows notable inaccuracies, leading to performance degradation as demonstrated in Table 2 and Table 3 of our paper.
>
> We also evaluated the end-to-end performance using H2O and SnapKV with chunked prefill on a subset of InfiniteBench:
>
> | Method | R.Number | E.Sum | E.MC | C.Debug |
> | - | - | - | - | - |
> | H2O | 3.39 | 15.35 | 45.41 | 20.57 |
> | SnapKV | 2.54 | 15.44 | 41.92 | 21.43 |
> | **Locret** | **97.46** | **16.82** | **46.29** | **29.71** |
>
> The results demonstrate that discrepancies between local and global importance scores in H2O and SnapKV lead to severe performance drops, particularly in R.Number. Locret, however, avoids such inconsistencies and achieves superior performance.
>
> We appreciate this insightful comment and will include this analysis and visualized results in our next revision.
>
> ---
>
> ### **W3: This paper lacks a detailed analysis of the CIS.**
>
> We have added further clarification regarding the purpose of designing CIS in our response to W2. Additionally, we will incorporate the discussion and experimental results outlined in W2 into the main text in our next revision.

---

> ### Author Response · Authors · 2024-11-15
> **Response to Reviewer xs3N (Part 2/3)**
>
> ### **Q1: Explaination of "the weakening correlation between local and global importance as sequences grow exacerbates this issue".**
>
> Existing cache importance scoring functions, such as H2O and SnapKV, are designed to identify important cache units only after the entire input sequence is prefilled. H2O relies on the complete attention scores to determine heavy hitters, while SnapKV’s voting mechanism requires the attention scores of the local window at the end of the input sequence, which also mandates full sequence prefill before eviction. When using chunked prefill, subsequent cache units that have not yet been processed are inaccessible, leading to significant discrepancies when applying H2O or SnapKV to prefilled cache units. This discrepancy arises because the predicted importance based on partial input diverges from the actual importance computed with the full sequence.
>
> We demonstrated this effect through an additional experiment in W2, which highlights the inconsistency in H2O and SnapKV. We hope this experiment clarifies the limitations of these methods in scenarios involving chunked prefill.
>
> ---
>
> ### **Q2: How effective would it be to directly use the maximum value of each column as a metric during inference?**
>
> We appreciate this question. However, it is not feasible to use the maximum value of each column as a metric during inference. As discussed in W2, a key objective of this paper is to integrate an eviction policy with chunked prefill. In chunked prefill, the subsequent cache units (tokens or hidden states) are not accessible, preventing us from calculating the attention score of a token in relation to all subsequent tokens. The maximum value of each column represents the highest attention score of a token’s query to the keys of all subsequent tokens, which cannot be determined during chunked prefill. Therefore, using the maximum value of each column as a metric during inference is not possible.
>
> ---
>
> ### **Q3: What are the actual performances of H2O combined with quantization?**
>
> Quantization combined with H2O leads to an attention shift, resulting in inaccurate cache importance estimation, as demonstrated in the Q-Hitters paper[4]. In Section 4.2, the authors report that the overlap ratio of identified heavy-hitters drops below 50% when quantization is applied. Additionally, Figure 7 of the Q-Hitters paper illustrates significant performance degradation when H2O is used with standard quantization techniques. We hope this addresses your question and clarifies the limitations of using H2O with quantization.
>
> ---
>
> ### **Q4: Comparison between SnapKV and PyramidKV.**
>
> Thank you for pointing this out. We will include H2O and SnapKV as additional baselines and have conducted comparative experiments with Locret on specific subsets of InfiniteBench. Since PyramidKV primarily manages budget allocation, it operates orthogonally to the eviction function and can be combined with Locret. We also provide results for the combination of Locret and PyramidKV. Due to the time-consuming nature of running the full benchmark, we were unable to generate complete results for InfiniteBench at this stage, but these will be included in our final revision.
>
> Additionally, the slow inference speed of H2O and SnapKV stems from their incompatibility with flash-attention, as both methods require access to the attention scores, which the current implementation of flash-attention does not support.
>
>
> | | R.Number | E.Sum | E.MC | C.Debug | Ave. |
> | - | - | - | - | - | - |
> | H2O | 3.39 | 15.35 | 45.41 | 20.57 | 21.18 |
> | SnapKV | 2.54 | 15.44 | 41.92 | 21.43 | 20.33 |
> | Locret | 97.46 | **16.82** | 46.29 | 29.71 | 47.57 |
> | Locret + PyramidKV | **99.66** | 15.82 | **48.03** | **30.00** | **48.38** |
>
> In our experiments, we modified only the scoring function, keeping all other hyperparameters consistent with Appendix A.2. When integrating with PyramidKV, we used maximum pooling among the CIS (following PyramidKV's setting) and set $\beta=2$.
>
> The results indicate that Locret outperforms H2O and SnapKV in chunked prefill scenarios for long-context inference. H2O and SnapKV show limitations in accurately predicting context retrieval tasks, such as R.Number. Additionally, incorporating PyramidKV for budget allocation management further enhances overall performance, demonstrating the compatibility between Locret and PyramidKV.

---

> ### Author Response · Authors · 2024-11-15
> **Response to Reviewer xs3N (Part 3/3)**
>
> ### **Q5: Is the Stabilizer used only for selecting recent tokens?**
>
> No, the stabilizers refer to the last $n_s$ tokens in each chunk during the chunked prefill process. These tokens are retained without eviction to maintain a local and continuous context, thereby minimizing errors (as stated in line 292).
>
> ---
>
> ### **Q6: Is the performance improvement in this paper due to the SFT? What would be the effect if SFT were directly applied to the model?**
>
> The performance improvement is not attributed to SFT. We use a minimal amount (3,000 entries) of long-context SFT data solely to train the retaining heads. Importantly, there is no SFT loss involved in Equation 3, and the LLM backbone remains frozen throughout. In other words, the only learnable component in our framework is the scoring function responsible for identifying which cache units are more important.
>
> ---
>
> ### **Q7: Should the number of heads in a retaining head be the same as in Query, or should it match Key/Value?**
>
> The number of retaining heads must match the number of Key/Value heads. To train the retaining heads, we select the **maximum** attention score (before softmax) **across different query heads within the same group** (as described in line 236).
>
> ---
>
> [1] H2O: Heavy-hitter oracle for efficient generative inference of large language models
>
> [2] SnapKV: LLM knows what you are looking for before generation
>
> [3] SirLLM: Streaming infinite retentive LLM
>
> [4] Q-Hitter: A Better Token Oracle for Efficient LLM Inference via Sparse-Quantized KV Cache

---

> ### Author Response · Authors · 2024-11-19
> **Manuscript updates**
>
> Manuscript updates are as follows. **Note: we use the new figure/table numbering here.**
>
> W1: line 841-842.
>
> W2, W3: Figure 1, Table 5.
>
> Q1: line 076-084.
>
> Q4: Table 5, Table 11.

---

> ### Author Response · Authors · 2024-11-25
> **Looking forward to futher discussions with Reviewer xs3N**
>
> Dear Reviewer xs3N,
>
> We would like to appreciate our gratefulness again for your valuable reviews. We have already uploaded the revised version based on your suggestions. Since the discussion period will be ending soon, we are looking forward to discussing the newly added content and our response with you. Should our responses have addressed your concerns, we would be grateful for an improved score. Thanks again for your time and effort.
>
> Submission 5591's Authors

---

> > ### Author Response · Authors · 2024-12-02
> > **Looking forward to futher discussions with Reviewer xs3N**
> >
> > Dear Reviewer xs3N,
> >
> > We sincerely thank you for you thoroughful review. We kindly remind you that the discussion phase will be ending in two days. We have provided some responses to the issues and questions raised in the review, and we have uploaded the manuscript with the additional experiments added and some parts rewritten. We are eager to have an extended discussion with you about the newly added content. Thanks again for your time and effort, and we wish you a happy holiday and New Year.
> >
> > Submission 5591 "Locret: Enhancing Eviction in Long-Context LLM Inference with Trained Retaining Heads" 's authors.

---

### Official Review · Reviewer_V3dC · 2024-11-02

**Soundness:** 2
**Presentation:** 3
**Contribution:** 3
**Rating:** 8
**Confidence:** 4

**Summary:**

The paper proposes LOCRET, an framework designed to enhance memory efficiency in long-context large language model (LLM) inference by using retaining heads to score and selectively retain key-value (KV) cache units. The primary challenge addressed is the high computational and memory demands posed by long-context LLM inference, which often limits deployment on consumer-grade devices. LOCRET introduces a trained retaining head mechanism that evaluates and prioritizes cache units based on their causal importance, offering a scalable and efficient approach that maintains inference quality on devices such as Nvidia 4090 GPUs. The paper conducts a comprehensive evaluation, comparing LOCRET with various memory-efficient inference baselines, demonstrating notable improvements in memory compression and inference quality without sacrificing speed.

**Strengths:**

1. The paper presents a framework combining trained retaining heads with chunked prefill, contributing a distinctive approach to KV cache management in long-context inference. Unlike previous methods, LOCRET’s retaining heads learn a heuristic for cache importance, adapting to specific model architectures and sequence types, which provides greater flexibility across transformer-based LLMs.
2. The empirical evaluation is rigorous, with comparisons across a diverse set of baselines, including INFLLM, Quantization, SIRLLM, and MINFERENCE. The experiments cover both long and shorter context scenarios, supporting the paper’s claims of LOCRET’s superiority in maintaining performance while reducing memory usage.
3. LOCRET offers a good solution for deploying long-context LLM inference on consumer-grade hardware by significantly reducing the KV cache size without compromising quality. This contribution is valuable given the rising importance of long-context LLM applications in various fields.
4. The paper is well-organized, providing a clear explanation of LOCRET's architecture, training process, and the underlying intuition behind retaining heads. Diagrams effectively illustrate the framework and its mechanisms, enhancing reader understanding of the complex process of cache unit scoring and selective eviction.

**Weaknesses:**

1. While the use of retaining heads to score and retain cache units is a valuable idea, the approach may benefit from further differentiation from existing token-dropping and quantization-based methods. Some parts of the scoring approach appear to overlap with traditional token importance estimation techniques (e.g., heavy-hitter approaches). A more comprehensive analysis highlighting LOCRET’s distinctions from similar heuristics in cache management would strengthen the contribution.
2. The results indicate promising efficiency gains but lack granular performance data on how LOCRET’s accuracy scales with different cache budgets across various architectures. Additionally, while the framework shows reduced memory requirements, further evidence on latency and computation trade-offs associated with retaining heads would be beneficial for practitioners evaluating deployment feasibility.
3. Although LOCRET is tested across two LLM architectures, the applicability of this approach to a broader set of LLMs with diverse attention mechanisms (e.g., sparse attention) is not explored in depth. Discussing potential limitations or adjustments required for alternative models would enhance the generalizability of the method.

**Questions:**

1. Could the authors clarify how LOCRET’s retaining heads would handle extremely high-context lengths (e.g., 10 million tokens)? Would additional constraints or modifications be required to manage the scoring of cache units in such contexts?
2. While SIRLLM performs poorly on memory-demanding tasks, it performs well on comprehension tasks. Could the authors comment on potential reasons LOCRET outperforms SIRLLM in these scenarios, particularly when both approaches manage memory through cache eviction?
3. Could the authors provide further insights or examples where the heuristic scoring might diverge significantly from the true causal importance? This would clarify the potential trade-offs in LOCRET's eviction policy.

---

> ### Author Response · Authors · 2024-11-16
> **Response to Reviewer V3dC (Part 1/3)**
>
> We sincerely appreciate your detailed review and valuable feedback.
>
> ---
>
> ### **W1: A more comprehensive analysis highlighting LOCRET’s distinctions from similar heuristics in cache management would strengthen the contribution.**
>
> Thanks for pointing this out. We have conducted the following experiment to highlight the differences between Locret and existing cache importance scoring functions, e.g. H2O and SnapKV. Cache importance scoring functions can generally be categorized into two types: causal and non-causal.
> - Non-causal scoring functions: Examples include H2O and SnapKV. These methods require information from subsequent cache units to determine the importance score of a cache unit, making them dependent on prefilling the entire sequence.
> - Causal scoring functions: Examples include SirLLM and our proposed method, Locret. These methods predict cache importance without relying on subsequent information.
>
> Non-causal scoring functions are incompatible with chunked prefill because they cannot calculate scores without access to the full sequence. If such functions are integrated with chunked prefill, they often face a significant discrepancy between the local importance score (without considering subsequent information) and the global importance score (with full context).
>
> To investigate this discrepancy, we measured the consistency of the top 10% most important cache positions identified in prefixes of various lengths compared to the full context. For reference, the full context is truncated to 6K tokens. The results are as follows:
>
> | Prefix Length | 0.5K | 1K | 1.5K | 2K | 2.5K | 3K | 3.5K | 4K | 4.5K | 5K | 5.5K | 6K (full) |
> | - | - | - | - | - | - | - | - | - | - | - | - | - |
> | H2O[1] | 75.46 | 80.58 | 83.30 | 85.00 | 87.20 | 89.22 | 90.76 | 92.48 | 94.02 | 95.54 | 97.40 | 100.00 |
> | SnapKV[2] | 37.12 | 31.17 | 29.20 | 32.09 | 29.60 | 32.71 | 28.80 | 29.20 | 31.82 | 34.86 | 41.65 | 100.00 |
> | SirLLM[3] | 100.00 | 100.00 | 100.00 | 100.00 | 100.00 | 100.00 | 100.00 | 100.00 | 100.00 | 100.00 | 100.00 | 100.00 |
> | Locret | 100.00 | 100.00 | 100.00 | 100.00 | 100.00 | 100.00 | 100.00 | 100.00 | 100.00 | 100.00 | 100.00 | 100.00 |
>
> This experiment highlights that scoring functions requiring future information, such as H2O and SnapKV, suffer from significant discrepancies when subsequent cache units are not considered. On the other hand, SirLLM, while also causal, shows notable inaccuracies, leading to performance degradation as demonstrated in Table 2 and Table 3 of our paper.
>
> We also evaluated the end-to-end performance using H2O and SnapKV with chunked prefill on a subset of InfiniteBench:
>
> | Method | R.Number | E.Sum | E.MC | C.Debug |
> | - | - | - | - | - |
> | H2O | 3.39 | 15.35 | 45.41 | 20.57 |
> | SnapKV | 2.54 | 15.44 | 41.92 | 21.43 |
> | **Locret** | **97.46** | **16.82** | **46.29** | **29.71** |
>
> The results demonstrate that discrepancies between local and global importance scores in H2O and SnapKV lead to severe performance drops, particularly in R.Number. It is this discrepancy that leads to the failure of H2O and SnapKV in accurately retrieving information from the context. Specifically, the model is unable to identify the importance of certain cache units at the time they are first encountered. Locret, however, avoids such inconsistencies and achieves superior performance.
>
> ---
>
> ### **W2.1: The results indicate promising efficiency gains but lack granular performance data on how LOCRET’s accuracy scales with different cache budgets across various architectures.**
>
> In order to investigate the impact of cache budget, we conducted experiments with different cache budgets on subsets of LongBench [4], using Phi-3-mini-128K with retaining heads trained on LongAlign. The results indicate that a larger cache budget generally leads to better performance, although certain tasks are more sensitive to cache budget variations than others.
>
> | Cache budget | 1024 | 2048 | 3072 | 4096 |
> | - | - | - | - | - |
> | GovReport | 27.94 | 30.96 | 31.72 | 32.72 |
> | MultifieldQA-En | 31.63 | 40.44 | 46.95 | 46.2 |
> | PassageRetrieval-En | 31.5 | 60.0 | 69.0 | 75.5 |
> | Dureader | 19.09 | 20.65 | 20.82 | 22.52 |
>
> We have tested Locret on two LLMs—Phi-3-mini-128K with an MHA architecture and Llama-3.1-8B-128K with a GQA architecture—demonstrating its compatibility with some of the most widely adopted model architectures. Exploring its applicability to other architectures, such as Encoder-Decoder models or MLA models, would be an interesting direction for future work. We will highlight this as part of the limitations in our next revision.

---

> ### Author Response · Authors · 2024-11-16
> **Response to Reviewer V3dC (Part 2/3)**
>
> ### **W2.2: Additionally, while the framework shows reduced memory requirements, further evidence on latency and computation trade-offs associated with retaining heads would be beneficial for practitioners evaluating deployment feasibility.**
>
> We have evaluated the model's forward throughput under varying context lengths, both with and without retaining heads. The results are summarized below. "R" represents the retaining heads, and the throughput is reported in tokens per second (tok/s) in the format "Ave. / Std."
>
> | Context Length | 1024          | 2048          | 3072          | 4096          |
> |----------------|---------------|---------------|---------------|---------------|
> | w/o R Speed   | 18674 / 443  | 19743 / 464  | 19982 / 402  | 20304 / 187  |
> | w/ R Speed    | 17118 / 1117| 18503 / 546  | 19054 / 283  | 19153 / 174  |
>
> From the results, no significant latency increase is observed when using retaining heads. The numerical differences are attributed to systematic variations rather than additional overhead introduced by retaining heads during inference.
>
> We have highlighted in our paper that the retaining heads trained for Phi-3-mini-128K and Llama-3.1-8B-instruct account for only 8% and 2.5% of the original model size, respectively (line 343). This minimal size overhead introduces negligible difficulty for deployment.
>
> ---
>
> ### **W3: Discussing potential limitations or adjustments required for alternative models would enhance the generalizability of the method.**
>
> Thanks for pointing this out. We will revise our limitation section to clearly state that Locret has been tested on decoder-only MHA and GQA architectures. Exploring the compatibility of Locret with other model architectures, such as MLA, remains part of our future work. Additionally, we aim to investigate the integration of Locret with other models, such as encoder-decoder architectures, in the future. This discussion will be incorporated into Section 5 (Discussion) in our next revision.
>
> ---
>
> ### **Q1: Could the authors clarify how LOCRET’s retaining heads would handle extremely high-context lengths (e.g., 10 million tokens)? Would additional constraints or modifications be required to manage the scoring of cache units in such contexts?**
>
> There are no additional constraints or modifications required for Locret to handle longer contexts.
>
> We conducted the following experiment to demonstrate Locret's capability in extremely long-context scenarios. We created a dataset similar to InfiniteBench's R.Number, with an average length of 10 million tokens. Each data point contains a 10-digit number string inserted into an irrelevant context, and the task is to retrieve the inserted number. The dataset consists of 50 examples, with the number strings uniformly distributed throughout the context. We used the hyperparameters from Table 6, with the exception of setting the chunk size to 10240 to speed up inference. The results, presented below, show that Locret can efficiently process extremely long contexts. In this experiment, the cache budget is set to 6000, and the compression ratio is 1747.6×.
>
> | Task | R.Number_10M |
> | - | - |
> | Acc. | 100.00 |
>
> ---
>
> ### **Q2: What are the potential reasons of Locret outperforming SirLLM on memory-demanding tasks?**
>
> There are two reasons why Locret outperforms SirLLM on memory-demanding tasks:
>
> First, SirLLM uses token-entropy to estimate token importance, where higher token-entropy indicates a token is harder to predict based on its context, deeming it more important. However, as highlighted in the Limitation Section of SirLLM's paper, significant discrepancies between user data and the model's training data can lead to poor eviction decisions. Memory-demanding tasks, such as context retrieval, often fall into such scenarios. For instance, tasks like R.PassKey and R.Number involve data patterns that are rare in natural contexts. As these patterns are not well-represented in the model’s training data, SirLLM fails to provide accurate token-entropy predictions, resulting in suboptimal performance.
>
> Second, SirLLM’s token-entropy is a token-level metric, while Locret's Causal Importance Score (CIS) operates at the cache unit level. This distinction allows Locret to assign different eviction policies across heads, enabling more flexible and effective cache management. In contrast, SirLLM lacks this flexibility, as its eviction strategy is uniformly applied. Previous studies [5, 6] have demonstrated that eviction patterns often vary among heads, and Locret’s ability to accommodate such variations provides a significant advantage in memory-demanding tasks.

---

> ### Author Response · Authors · 2024-11-16
> **Response to Reviewer V3dC (Part 3/3)**
>
> ### **Q3: Could the authors provide further insights or examples where the heuristic scoring might diverge significantly from the true causal importance?**
>
> Since Locret is a training-based method, it may perform poorly on contexts that have low probability in the training data. For instance, punctuation is often treated as unimportant in the training dataset. If the query focuses on retrieving specific punctuation marks from a long context, Locret may provide a biased CIS and incorrectly evict important cache units.
>
> However, this issue can be mitigated by designing a more representative training dataset. Moreover, such problems are inherently challenging even for humans to solve, as they require understanding the context where insignificant elements like punctuation become important. We will address this limitation in our next revision.
>
> ---
>
> [1] H2O: Heavy-hitter oracle for efficient generative inference of large language models
>
> [2] SnapKV: SnapKV: LLM knows what you are looking for before generation
>
> [3] SirLLM: Streaming infinite retentive LLM
>
> [4] Longbench: A bilingual, multitask benchmark for long context understanding
>
> [5] Model Tells You What to Discard: Adaptive KV Cache Compression for LLMs
>
> [6] MInference 1.0: Accelerating Pre-filling for Long-Context LLMs via Dynamic Sparse Attention

---

> ### Author Response · Authors · 2024-11-19
> **Manuscript updates**
>
> Manuscript updates are as follows. **Note: we use the new figure/table numbering here.**
>
> W1: Figure 1, Table 5.
>
> W2: Figure 6(a), Table 18.
>
> W3, Q3: line 076-084.
>
> Q1: Table 16.

---

### Author Response · Authors · 2024-11-19
**Revised Manuscript updated by Authors**

We are very grateful to the reviewers for their detailed and thoughtful comments! We have responded to each comment and conducted additional experiments and clarifications. Based on the reviewers' feedback, we have updated and uploaded the revised manuscript.

**We use blue color for all newly added content.** We will turn them back to black in the final version. Next, we will provide a one-on-one response to each reviewer, stating the specific locations in the paper where modifications have been made.

Due to the addition of new figures and tables, the numbering in the revised manuscript differs from the initial version. We provide a table comparing the old and new figure/table numbers for the reviewers' convenience when reading our response. In the response, we use the old numbering.

### **Figures**

| Old numbering | New numbering |
| :-: | :-: |
| Figure 1 | Figure 2 |
| Figure 2 | Figure 3 |
| Figure 3 | Figure 4 |
| Figure 4 | Figure 5 |
| Figure 5 | Figure 7 |
| Figure 6 | Figure 8 |
| Figure 8 | Figure 9 |

### **Tables**


| Old numbering | New numbering |
| :-: | :-: |
| Table 1 | Table 1 |
| Table 2 | Table 3 |
| Table 3 | Table 4 |
| Table 4 | Table 9 |
| Table 5 | Table 10 |
| Table 6 | Table 2 |
| Table 7 | Table 6 |
| Table 8 | Table 12 |
| Table 9 | Table 13 |
| Table 10 | Table 14 |
| Table 11 | Table 15 |

---

### Comment · Area_Chair_m9ij · 2024-11-23
**Engage in Discussions Before Nov 26 (AoE)**

Dear Reviewers,

First, let me thank you for your invaluable contributions to the ICLR review process. Your constructive feedback plays a key role in enhancing the quality of submissions.

---

As we approach the final days of the discussion phase (ending **Nov 26, 2024, AoE**), I kindly remind you to:

- Please take a moment to review the authors' responses to your comments. This is an opportunity to clarify any remaining questions, acknowledge misunderstandings, and refine your evaluation.

- If you need further clarification, don't hesitate to post your comments as soon as possible.

- If the authors' responses address your concerns or provide new insights, please consider updating your score to reflect this.

---

Your thoughtful participation during this phase is especially valuable for borderline papers, where additional input can be critical to ensuring a fair decision-making process.

I understand how busy this time of year can be and truly appreciate the time and care you dedicate to this important role. Your efforts make a tangible impact on the success of ICLR.

Thank you once again for your dedication.

Best regards,

Area Chair, ICLR 2025

---

### Meta-Review · Area_Chair_m9ij · 2024-12-25

**Metareview:**

The paper introduces LOCRET, a framework for memory-efficient long-context inference in LLMs. LOCRET employs lightweight retaining heads trained to predict the causal importance of KV cache units, enabling selective cache eviction. Using this mechanism with `chunked prefill`, LOCRET achieves efficient inference on consumer-grade GPUS while maintaining strong performance on long-context tasks.

## Strengths

- LOCRET introduces a query-independent mechanism for cache eviction, which addresses the bottleneck of KV cache growth in long-context inference (Reviewer WKXD). The paper enables long-context inference on consumer-grade hardware.
- The paper includes detailed comparisons across multiple benchmarks and baselines (e.g., SnapKV, SirLLM, and H2O) and explores various hyperparameters like cache budget, stabilizer length, and chunk size (Reviewer WKXD, Reviewer V3dC).
- LOCRET consistently demonstrates strong memory efficiency and performance on a variety of tasks (Reviewer V3dC).
- The framework requires minimal training resources (less than 1 GPU hour) and makes only minor modifications to existing LLM architectures, ensuring diverse applicability (Reviewer xs3N, Reviewer WKXD).

## Weaknesses

- LOCRET has been evaluated primarily on decoder-only architectures and does not generalize to encoder-decoder models or alternative attention mechanisms, such as sparse attention (Reviewer WKXD, Reviewer j7cb).
- Poor performance on complex query-driven tasks like RULER indicates that LOCRET struggles without query-awareness, limiting its applicability in certain retrieval scenarios (Reviewer KmJZ, Reviewer xs3N).
- While the experiments are extensive, some reviewers noted a lack of focus on challenging benchmarks or real-world datasets (Reviewer KmJZ). For example, InfiniteBench’s simplicity and the lack of off-topic turns in multi-turn conversation datasets reduce the impact of the results.
- SnapKV and similar baselines were tested under chunked prefill scenarios, which are not compatible with their design, potentially inflating LOCRET’s relative performance (Reviewer KmJZ).
- LOCRET exhibits performance degradation on complex retrieval tasks like multi-key RULER subtasks, raising concerns about its robustness and generalizability (Reviewer KmJZ).

Based on the reviews and the rebuttal, I recommend **Reject** for this paper. The paper tackles an important problem, however, the paper in its current form has few issues that seem not to be properly addressed in the rebuttal:

- LOCRET fails to generalize beyond decoder-only architectures, struggling with encoder-decoder models and alternative attention mechanisms (Reviewer WKXD, Reviewer j7cb).
- Its poor performance on query-driven benchmarks like RULER, especially on complex multi-key tasks, raises concerns about its applicability in real-world scenarios (Reviewer KmJZ). The results suggest that LOCRET lacks the generalizability needed to handle tasks where query-awareness is critical.
- While the authors run extensive evaluations, some of the chosen benchmarks (e.g., InfiniteBench) are overly simplistic and fail to challenge the proposed method sufficiently (Reviewer KmJZ).
- The comparisons with baselines like SnapKV under chunked prefill scenarios are unfair, as these methods are not designed for such setups (Reviewer KmJZ).
- The proposed method, while practical, does not demonstrate significant contribution over existing approaches like SnapKV and H2O. The core idea of training retaining heads to predict cache importance lacks sufficient theoretical development and differentiation from prior work (Reviewer xs3N, Reviewer WKXD).
- The paper does not convincingly address challenges in query-driven or multi-turn conversation tasks. LOCRET struggles with off-topic or inconsistent input contexts, and the experiments lack robust evaluation in such scenarios (Reviewer KmJZ).

**Additional Comments On Reviewer Discussion:**

During the rebuttal period, the authors actively engaged with the reviewers and addressed many of their concerns by running additional experiments and clarifying key aspects of the paper. However, certain critical issues remained unresolved.

- (Reviewer V3dC) emphasized the need for better differentiation between LOCRET and existing methods like SnapKV and H2O, as well as more granular performance analysis on varying cache budgets and architectures. While the authors made an effort to address these points, the reliance on chunked prefill continued to be a limitation, as noted by other reviewers. The additional results strengthened the empirical validation but did not resolve the fundamental concerns regarding fair comparisons.

- (Reviewer xs3N) raised concerns about the modest novelty of LOCRET, the effectiveness of the causal importance score (CIS), and the reliance on additional training. Despite the detailed response, the concerns about the limited novelty persisted. The additional experiments highlighted LOCRET's benefits but did not fully address the critique of its generalizability or query-unaware limitations.

- (Reviewer j7cb) limited evaluation on diverse LLM architectures and benchmarks. The reviewer sought validation on additional models and datasets like LongBench. The new experiments partially addressed the reviewer’s concerns, demonstrating LOCRET’s applicability to other LLMs. However, the broader limitation of not evaluating on encoder-decoder architectures or other attention mechanisms remained unaddressed.

- (Reviewer KmJZ) critiqued the reliance on simplistic benchmarks like InfiniteBench, LOCRET’s weak performance on complex tasks in RULER, and unfair comparisons with SnapKV under chunked prefill. They recommended fair evaluation on shorter contexts (e.g., RULER 4K) without chunked prefill and highlighted LOCRET’s struggles with task complexity. I reached out to the reviewers to assess whether the rebuttal addressed their concerns. Reviewer KmJZ provided a very useful comment that reinforced my conclusion that the authors did not adequately address the concerns about LOCRET's practicality, fairness in baseline comparisons, and generalizability.

Overall, I reiterate that this is a good research direction, however, the paper needs to carefully address the reviewers' concerns (complex tasks, showing general applications on different architecture, fair baseline). Given the significant concerns raised by Reviewer KmJZ and the lack of satisfactory rebuttal, I recommended reject for the paper.

---

### Decision · Program_Chairs · 2025-01-22

Reject